# Deliberation Meets Reaction: A Dual-Expert VLA framework for Autonomous Driving

## Abstract

Vision-Language-Action (VLA) models have emerged as a promising paradigm for end-to-end autonomous driving due to their remarkable interpretability and generalization. However, their practical deployment is severely hindered by substantial computational costs and high inference latency. This challenge stems from (1) a large number of model parameters for maintaining world knowledge and (2) intensive Chain-of-Thought (CoT) reasoning for improving driving performance. Inspired by the observation that experienced drivers usually only engage in intensive deliberation in unfamiliar or complex situations, we propose an adaptive **d**ual-**e**xpert VLA model, termed **DE-Driver**, to adaptively select activated experts and reduce unnecessary reasoning. Specifically, DE-Driver integrates a lightweight reactive expert for swift responses and a powerful deliberative expert for complex reasoning. Depending on the scenario, a scene-aware router dynamically directs layer-wise features to the appropriate expert. Then, these selected experts determine whether to generate CoT reasoning, ensuring a balance between inference efficiency and driving performance. Experimental results on the closed-loop Bench2Drive benchmark show that DE-Driver achieves driving performance on par with state-of-the-art methods while significantly improving inference efficiency.

## 1 Introduction

Autonomous driving has evolved from a modularized paradigm (Zhang et al. (2022); Chitta et al. (2023)) to an end-to-end (E2E) paradigm (Hu et al. (2023); Jiang et al. (2023); Guo et al. (2025)) that allows joint optimization across all components and reduces cascading errors. However, classic end-to-end models are trained by imitating expert trajectories, leading to a lack of understanding about surrounding environments (Li et al. (2025a), Jiang et al. (2025)), resulting in poor performance in closed-loop Bench2Drive benchmark (Jia et al. (2024)). Due to extensive world knowledge and generalization from Large Language Models (LLMs), Vision-Language-Action (VLA) framework achieves remarkable success in autonomous driving and becomes the frontier paradigm. VLA frameworks bridge the semantic gap prevalent in long-tail scenarios and generate explicit Chain-of-Thought (CoT) to explain the reasons for their decision.

Although promising, current VLA models (Tian et al. (2024); Chen et al. (2024b); Renz et al. (2025); Yang et al. (2025); Zhou et al. (2025)) suffer from the substantial computational overhead of LLMs with billions of parameters, making it challenging to infer on resource-constrained onboard hardware. This inefficiency manifests in two key stages of inference: (1) during the prefill stage, the model must process the long sequence of visual and text tokens by large-scaled trained parameters, (2) during auto-regressive decoding, the answer with CoT and actions is generated token-by-token.

Most VLAs continuously generate long reasoning chains to ensure driving safety and effectiveness, resulting in a high amount of computation and time consumption, as shown in Figure 1 (b). In contrast, an experienced driver engages in fast reactive driving in common scenarios and slow deliberative driving in difficult scenarios, saving energy and improving efficiency, as shown in Figure 1 (a). **Therefore, this paper studies how to equip a VLA with two distinct driving modes and an adaptive switching mechanism, enabling a transition from a "novice driver" that deliberates everywhere to an "experienced driver" that thinks deeply only at appropriate times.**

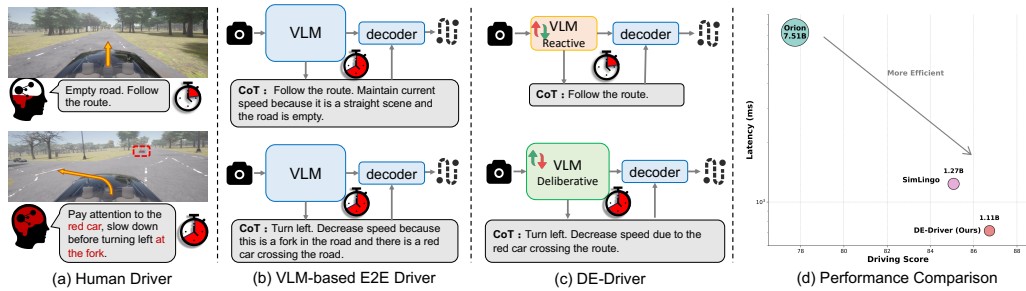

Figure 1: Comparison of Driving Strategies. (a) Human drivers excel at adapting their thinking state to the scene, allowing for swift responses while saving effort. (b) VLM-Based E2D Driver persistently engage in computationally intensive CoT reasoning, resulting in high inference latency. (c) DE-Driver adaptively switches between reactive and deliberative experts, thereby reducing inference latency. (d) Performance comparison demonstrates the effectiveness of DE-Driver.

To this end, we propose **DE-Driver**, a dual-expert Vision-Language-Action (VLA) model to dynamically adjust driving modes, as illustrated in Figure 1 (c). It features two key components: Dual-Expert (**DE**) module and Progressive Expert Specialization (**PES**) strategy. The DE module employs a heterogeneous architecture comprising a lightweight reactive expert for quick responses in simple scenarios and a powerful deliberative expert for handling complex events that require careful reasoning. This module utilizes a scene-aware router, governed by an expert load loss, to dynamically assign tokens to the most suitable expert at each layer. This mechanism enables the model to effectively learn contextual scene semantics and dynamically adjust the expert routing. Furthermore, the PES strategy is a three-stage training process designed to progressively specialize experts and their assignments. This process includes: (1) creating the reactive expert via structural pruning of a foundation model; (2) enhancing its capabilities through knowledge distillation from the deliberative expert; and (3) collaboratively finetuning both experts and the router on a mixed dataset comprising both a mixed dataset with and without CoT, which stabilizes the expert switch and reasoning chain generation. The results in Figure 1 (d) demonstrate the effectiveness of DE-Driver. The main contributions are summarized as follows:

- We propose DE-Driver, a dual-expert VLA framework aiming to adaptively active experts and reduce unnecessary reasoning to improve efficiency in autonomous driving.

- We design a Dual-Expert (DE) module that dynamically switches between a lightweight reactive expert and a powerful deliberative expert according to the semantics of the driving scenario, reducing the computational load during the prefilling stage.

- We introduce a Progressive Expert Specialization (PES) strategy to drive clear expert roles and stable routing training. Meanwhile, it enables the model to selectively skip the generation of unnecessary CoT, thereby accelerating the autoregressive decoding stage.

- Compared to existing methods based on the closed-loop Bench2Drive benchmark, DE-Driver achieves state-of-the-art performance with significantly reduced inference latency.

## 2 RELATED WORK

### 2.1 END-TO-END AUTONOMOUS DRIVING

Recent end-to-end autonomous driving research has primarily focused on enhancing planning capabilities through various architectural designs. Transformer-based frameworks (Hu et al. (2023), Jiang et al. (2023), Jia et al. (2023b), Jia et al. (2025)) facilitate the fusion of multi-modal inputs and the direct prediction of driving trajectory. Meanwhile, anchor-based methods (Chen et al. (2024a), Li et al. (2024), Song et al. (2025), Li et al. (2025b)) achieve multi-mode planning by predicting a probabilistic distribution of candidates, degrading the driving uncertainty. To improve the diversity of trajectory generation, diffusion-based methods (Liao et al. (2025), Wang et al. (2025b)), Zhao et al. (2025)) are introduced to model the underlying distribution space of future trajectories. However, these approaches still struggle with long-tail cases in a closed-loop driving environment.

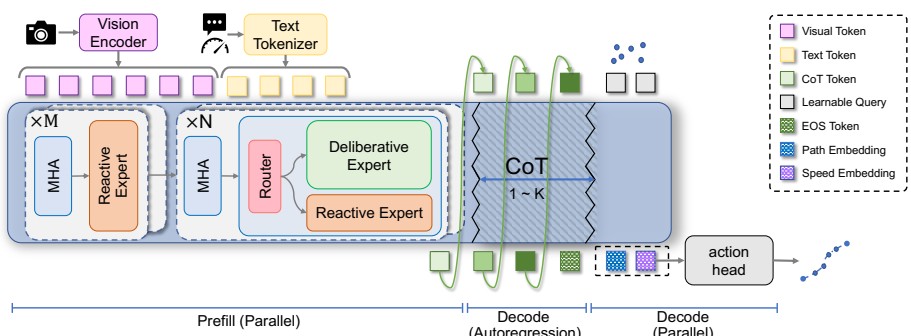

Figure 2: Overall Architecture of DE-Driver. It integrates lightweight reactive expert and powerful deliberative expert to handle different scenario, improving prefill time. Meanwhile, it adaptively reduce unnecessary CoT to reduce decode latency in autoregression stage.

## 2.2 VLM-based Autonomous Driving

Integrating Vision-Language Models (VLMs) into end-to-end autonomous driving systems has become a prominent research direction for addressing long-tail scenarios. Early works utilized VLMs as an auxiliary component to provide high-level guidance, such as reference trajectories (Tian et al. (2024)), meta-actions (Jiang et al. (2024)), or feature embeddings (Chen et al. (2024b); Liu et al. (2025)), thus reducing decision uncertainty in complex situations. However, the semantic gap between the VLM and other driving models often hindered both systems from realizing their full potential. To bridge this gap, subsequent research has employed VLM as the core of the driving system to directly generate outputs like textual trajectories (Li et al. (2025a); Wang et al. (2025a)) or discretized action tokens from a predefined codebook (Zhou et al. (2025)). Nevertheless, generating precise, continuous floating-point trajectories via natural language remains a significant challenge. To overcome this, a common paradigm is to append a dedicated action decoder, such as the diffusion-based planner (Guo et al. (2025); Li et al. (2025c)), MLP (Renz et al. (2025)), or VAE-GRU (Jiang et al. (2025)), to translate the high-level reasoning latent space of the VLM into low-level control actions. While those approaches have improved driving performance in long-tail scenes, they struggle with substantial computational overhead and inference latency, impeding practical deployment. Recent efforts to accelerate VLA systems have explored structured pruning (Cao et al. (2025)) and Mixture of Experts (MoE) (Zhang et al. (2024); Yang et al. (2025)). While recent works like Fast-DriveVLA (Cao et al. (2025)) efficiently reduce spatial redundancy by pruning visual tokens during the prefill stage, DE-Driver focuses on reducing computational and temporal redundancy during the autoregressive decoding stage by adaptively skipping CoT and routing experts. Consequently, instead of risking the loss of important visual cues via aggressive pruning, we focus on redesigning the VLA structure to reduce latency while fully preserving input features and improving the reasoning capability for end-to-end autonomous driving.

## 3 Methodology

To address this inherent trade-off between performance and efficiency, we propose Dual-Expert VLA framework, termed DE-Driver as shown in Figure 2. It contains two core components of DE-Driver: (1) Dual-Expert (DE) module, the key to enabling conditional computation; and (2) Progressive Expert Specialization (PES) strategy, a novel training paradigm to ensure functional differentiation of experts and precise routing.

### 3.1 Overall Architecture

DE-Driver is built upon SimLingo (Renz et al. (2025)). Its central innovation is the replacement of conventional Feed-Forward Network (FFN) layers with our proposed dual-expert module.

**Multimodal Inputs:** The model uses image $\mathcal{I} \in \mathbb{R}^{H \times W \times 3}$ and command $\mathcal{C} \in \mathbb{R}^{N_t}$ as inputs. In particular, we use a vision encoder $E_v$ (*e.g.* ViT (Dosovitskiy et al. (2021)) and a text tokenizer $E_c$

to map $\mathcal{I}$ and $\mathcal{C}$ to a set of tokens $\mathbf{Z}_v \in \mathbb{R}^{N_v \times d}$ and $\mathbf{Z}_t \in \mathbb{R}^{N_t \times d}$, respectively, where $d$ is the hidden dimension of the model.

$$\mathbf{Z}_v = E_v(\mathcal{I}), \quad \mathbf{Z}_c = E_c(\mathcal{C})$$
$$\mathbf{H}_{v,c}^{(0)} = \text{Concat}(\mathbf{Z}_v, \mathbf{Z}_c) \tag{1}$$

**LLM with Dual-Experts:** The concatenated tokens $\mathbf{H}$ are fed into a decoder-only Transformer backbone $f(\theta)$ to generate CoT text and action embedding. During the prefill stage, the model processes all input tokens in parallel to fuse multimodal information and comprehend the driving context. Its distinction lies in replacing some of the standard Transformer blocks' FFNs with our Dual-Expert (DE) modules. As shown in Figure 3 (a), our $\text{LLM}_{\text{DE}}$ consists of N Transformer blocks with reactive expert and final M blocks incorporating our MoE layers, enabling the model to adaptively select computational paths for each token at these layers.

$$\mathbf{H}^{(l)} = \begin{cases} \text{ReactiveExpert}(\mathbf{H}^{(l-1)}) & \text{for } 1 \leq l \leq L - M \\ \text{DualExpert}(\mathbf{H}^{(l-1)}) & \text{for } L - M + 1 \leq l \leq L \end{cases} \tag{2}$$

**CoT and Action Decoding:** Following the prefill stage, the model enters an autoregressive decoding phase to generate an optional CoT and the final action $\mathcal{A}$. Our CoT include tow parts: control signal (*e.g.,* 'turn left') and behavioral explanation (*e.g.,* 'decelerate due to the child intersecting your path'). Because the token length of control signal is significantly smaller than that of behavioral explanation, our method continuously generates control signal tokens $\mathbf{Z}_{cot}^c$ while adaptively activating behavioral explanation only when necessary. If the scene is deemed complex, the model first generates a sequence of CoT tokens of behavioral explanation $\mathbf{Z}_{cot}^b = \{Z_{\text{cot\_1}}^b, .., Z_{\text{cot\_k}}^b\}$ for explicit reasoning,then action tokens $\mathbf{Z}_a$ are yielded conditioned on the full context ($\mathbf{Z}_v$, $\mathbf{Z}_t$, $\mathbf{Z}_{cot}^c$, $\mathbf{Z}_{cot}^b$). For simple scenes, DE-Driver only reasons control signal tokens and then predicts the action tokens $\mathbf{Z}_a$. Finally, the action tokens $\mathbf{Z}_a$ are passed to an Action Head $\text{D}_a$(*e.g.,* MLP (Renz et al. (2025)) to regress the final driving waypoints $\tau$.

$$\mathbf{Z}_{\text{cot}}^c = \text{LLM}_{\text{DE}}(\mathbf{Z}_v, \mathbf{Z}_t), \quad \mathbf{Z}_{\text{cot}}^b = \text{LLM}_{\text{DE}}(\mathbf{Z}_v, \mathbf{Z}_t, \mathbf{Z}_{\text{cot}}^c),$$
$$\mathbf{Z}_a = \text{LLM}_{\text{DE}}(\mathbf{Z}_v, \mathbf{Z}_t, \mathbf{Z}_{\text{cot}}^c, \mathbf{Z}_{\text{cot}}^b), \quad \tau = \text{D}_a(\mathbf{Z}_a) \tag{3}$$

## 3.2 DUAL-EXPERT MODULE

This module is the core of DE-Driver architecture. It comprises a router and two functionally specialized expert networks that make a computational choice for each token at every MoE layer.

**Expert Construction:** We adopt a standard FFN of LLMs as deliberative expert. It maintains full world knowledge learned from large web data. Its activation is strongly correlated with subsequent CoT generation, providing an interpretable basis for the model's decisions. Meanwhile, we restruct a lightweight FFN as Reactive Expert by simply reducing the half intermediate dimensionality of deliberative expert. It impairs the large model's extrapolation capabilities to some extent, but can still learn simple control behaviors.

**Scene-Aware Router:** Each transformer layer of LLM can facilitate the interaction between visual tokens and other tokens, resulting in various scene semantic. Therefore, we design a scene-aware router $\mathcal{G}$, a learnable linear layer, to assess scene difficulty and yield expert scores using the visual evidence available at the current layer. Specially, our routing is layer-wise rather than token-wise, at each Transformer layer we select a single expert, and that expert processes all tokens in that layer. Concretely, mean-pooling and max-pooling are used to integrate global and notable scene features $r^{(l)} \in \mathbb{R}^{2d}$ over all visual tokens of the current layer. Then, the router $\mathcal{G}$ calculate scores $p \in \mathbb{R}^2$ and select the most suitable expert $e^{(l)} \in \{0, 1\}$ to process all tokens at layer $l$.

$$\mathbf{h}_r^{(l)} = \text{Concat}(\text{MaxPool}(\mathbf{H}_v^{(l)}), \text{MeanPool}(\mathbf{H}_v^{(l)}))$$
$$e^{(l)} = \arg\max(p^{(l)}), \quad p^{(l)} = \mathcal{G}^{(l)}(\mathbf{h}_r^{(l)}) \tag{4}$$

**Expert Load Loss:** Our key supervisory that explicitly links scene complexity to expert selection: simple scenes should rely more on reactive experts, whereas complex scenes should involve a higher fraction of deliberative experts. for this purpose, we derive a binary scene-difficulty label $y_s \in \{0, 1\}$

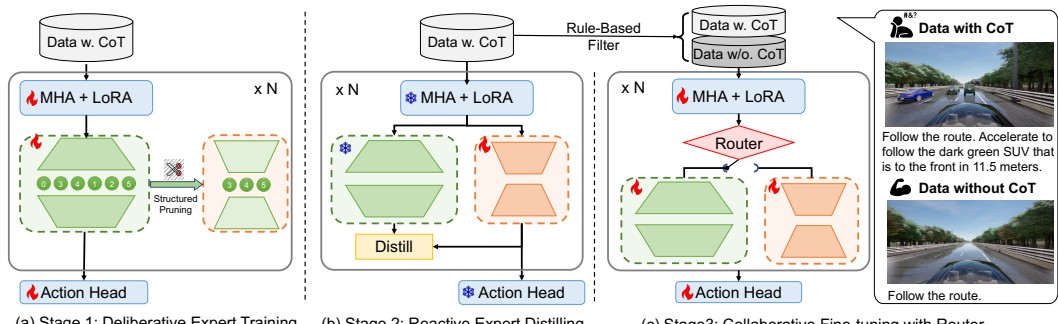

(a) Stage 1: Deliberative Expert Training    (b) Stage 2: Reactive Expert Distilling    (c) Stage3: Collaborative Fine-tuning with Router

Figure 3: Overview of PSE Strategy. Its three stages are responsible for obtain a basic reasoning ability, a powerful reactive expert , and adaptive VLA system, respectively.

using rule-based estimates of vehicle activeness, acceleration ranges, etc. In addition, we use agent proportion, $y_{\text{agent}}$, in image and CoT length proportion $y_{\text{text}} = \text{clip}(l/L_{\max}, 0, 1)$ as key supervisory that positively correlate with scene complexity.

$$l_{\text{text}} = \text{MSE}(\frac{1}{N}\sum e^{(l)}, y_{text}), \quad l_{\text{agent}} = \text{MSE}(\frac{1}{N}\sum(e^{(l)}), y_{\text{agent}}),$$

$$l_s = \text{CE}(\frac{1}{N}\sum_{l=1}^{N} p^{(l)}, y_s), \quad \mathcal{L}_r = l_s + \alpha_1 l_{\text{text}} + \alpha_2 l_{\text{agent}} \tag{5}$$

## 3.3 PROGRESSIVE EXPERT SPECIALIZATION STRATEGY

To train two experts and router stably, we propose the Progressive Expert Specialization strategy (PES) including a three training stage: deliberative expert training, reactive expert distillation, and collaborative fine-tuning with router, as illustrated in Figure 3.

**Deliberative Expert Training:** The initial stage of PES focuses on establishing a strong foundation for both experts. To imbue it with sophisticated reasoning abilities, we leverage rich CoT dataset $D_{CoT}$. The training objective is to maximize the likelihood of generating the correct reasoning chains and driving action. To maintain parameter efficiency, we employ Low-Rank Adaptation (LoRA) specifically on the Multi-Head Attention (MHA) layers of the model. Upon convergence of the deliberative expert, we generate a reactive expert by applying structured pruning with L2 importance score to the deliberative expert, which reserves Top-K important intermediate weights of FFN. This approach guarantees tangible reductions in latency and memory footprint on modern hardware. The training loss is the smooth L1 loss between predict trajectory $\hat{\tau}$ and ground truth $\tau$.

$$\mathcal{L}_\tau = \text{SmoothL1}(\hat{\tau}, \tau) \tag{6}$$

**Reactive Expert Distilling:** The reactive expert, despite inheriting weights from a powerful teacher, suffers from a substantial loss of semantic knowledge due to pruning. To recover its performance and refine its capabilities, we employ knowledge distillation. In this stage, the fully trained deliberative expert acts as a static, frozen teacher model, while the pruned reactive expert serves as the student model. We use the same CoT dataset ($D_{CoT}$) to provide input to both models. The core idea is to train the reactive expert to mimic the output distribution of the more capable deliberative expert. This forces the compact model to learn the nuanced decision-making logic of its larger counterpart. The distillation loss is MSE between the features produced by the teacher and the student:

$$\mathcal{L}_{Distill} = \sum_{l=1}^{N} \text{MSE}(H_D^{(l)}, H_R^{(l)}) \tag{7}$$

The training loss $\mathcal{L}_{stage2}$ for the reactive expert is a weighted sum of a task-specific loss on the ground-truth labels and the distillation loss that aligns the student's features with the teacher's.

$$\mathcal{L}_{\text{stage2}} = \mathcal{L}_\tau + \beta \mathcal{L}_{\text{Distill}} \tag{8}$$

**Collaborative Fine-tuning with Router:** The final stage aims to integrate the two specialized experts into a single, cohesive system governed by a lightweight router. The goal is to dynamically dispatch queries based on their inferred complexity, while simultaneously fine-tuning all components to work in concert. In this stage, we employ a rule-based approach to eliminate CoT data in simple scenarios as dataset $D_{NCoT}$. We unfreeze the parameters of our experts and action head to finetune the reasoning of different expert ratios under mixed dataset $D_{mix} = D_{\text{CoT}} \cup D_{\text{NCoT}}$. The training objective $\mathcal{L}_{\text{stage3}}$ consists of loss of a router and loss of tasks, as Eq (9). Here, $D_{mix}$ will be allocated the $\text{LLM}_{\text{DE}}$ to learn whether to generate $\mathbf{Z}_{cot}^{b}$ and the ratio of deliberative experts will be optimized.

$$\mathcal{L}_{\text{stage3}} = \mathcal{L}_{\tau} + \gamma \mathcal{L}_{r} \tag{9}$$

## 4 EXPERIMENTS

### 4.1 DATASET, EVALUATION METRICS AND IMPLEMENTATION DETAILS

**Dataset:** Our work is based on the extensive driving dataset introduced by SimLingo (Renz et al. (2025)), which was collected in the CARLA simulator using the privileged, rule-based expert, *PDM-lite* (Sima et al. (2024)). The dataset comprises approximately 3.1 million samples captured at 4 frames per second. The data collection spans a diverse set of driving conditions and scenarios across multiple route types. Besides, we restructured the length of the reasoning chain into a mixed dataset $D_{mix}$ using rules such as driving jerk, traffic sign, and surrounding vehicles.

**Evaluation Metrics:** For closed-loop evaluation, we employ the Bench2Drive benchmark (Jia et al. (2024)). This benchmark, running on CARLA 0.9.15, provides a standardized testbed consisting of 220 short routes (approximately 150m each) distributed across Towns 1 through 15 under a variety of weather conditions. We report performance using its official suite of metrics, which encompasses Driving Score (DS), Success Rate (SR), Efficiency, comfort, and Multi-Ability. Meanwhile, we test the latency of vision, action head, and LLM of the VLA system on the RTX4090 GPU.

**Implementation Details:** Our implementation largely follows the setup described in Sim-Lingo (Renz et al. (2025)). We maintain the same model architecture and training configuration to ensure a fair comparison. Specifically, we utilize AdamW optimizer (Loshchilov & Hutter (2019)) with a learning rate of 3e-5, a weight decay of 0.1, and a cosine annealing schedule. The model was trained on a cluster of 8 NVIDIA A100 (80GB) GPUs with a batch size of 12 and an epoch of 14. We leverage DeepSpeed v2 to optimize memory usage and training efficiency. We experientially set $M = 12$, $\alpha_1 = 0.5$, $\alpha_2 = 0.01$, and $\gamma = 0.1$.

### 4.2 MAIN RESULTS

**Driving Performance on Bench2Drive:** As reported in Table 1, DE-Driver establishes a new state-of-the-art on the Bench2Drive benchmark. Our model achieves a Driving Score (DS) of **86.73** and a Success Rate (SR) of **67.73%**, outperforming all existing end-to-end and VLA methods. Notably, DE-Driver surpasses our strong baseline, SimLingo (Renz et al. (2025)), demonstrating that our dual-expert framework not only improves efficiency but also enhances driving capability. Meanwhile, we examine the underlying reasons for the drop in driving comfort in Appendix A.6.

**Multi-Ability:** The fine-grained Multi-Ability results are presented in Table 2. DE-Driver achieves the highest mean score of **67.87%**, showcasing its proficiency across a wide range of complex scenarios. Specifically, our model excels in challenging interactive situations such as Merging (60.00%) and Give Way (60.00%), where it achieves the top performance. This suggests that our deliberative expert, guided by the scene-aware router, is effectively activated in these complex scenarios to perform more nuanced and safer maneuvers. While SimLingo shows slightly stronger results in tasks like Emergency Brake and Overtaking, our model remains highly competitive across all categories, proving its well-rounded and robust driving intelligence.

**Inference Efficiency:** The primary motivation behind DE-Driver is to mitigate the high computational cost of VLA models without compromising performance. The results in Table 3 convincingly validate our approach. Our model adopts an MoE architecture with a total of 1.43 billion parameters,

Table 1: Closed-Loop Results on the Bench2Drive Benchmark. * denotes expert feature distillation. **Bold** and underline denote the highest and second-highest scores, respectively.

| Method | Reference | DS | SR(%) | Efficiency | comfort |
|---|---|---|---|---|---|
| TCP-traj* ( Wu et al. (2022)) | NeurIPS 22 | 59.90 | 30.00 | 76.54 | 18.08 |
| AD-MLP ( Zhai et al. (2023)) | arXiv 23 | 18.05 | 0.00 | 48.45 | 22.63 |
| VAD ( Jiang et al. (2023)) | ICCV 23 | 42.35 | 15.00 | 157.94 | 46.01 |
| UniAD-Base ( Hu et al. (2023)) | CVPR 23 | 45.81 | 16.36 | 129.21 | 43.58 |
| ThinkTwice* ( Jia et al. (2023b)) | CVPR 23 | 62.44 | 31.23 | 69.33 | 16.22 |
| DriveAdapter* ( Jia et al. (2023a)) | ICCV 23 | 64.22 | 33.08 | 70.22 | 16.01 |
| GenAD ( Zheng et al. (2024)) | ECCV 24 | 44.81 | 15.90 | - | - |
| DriveTransformer ( Jia et al. (2025)) | ICLR 25 | 63.46 | 35.01 | 100.64 | 20.78 |
| MomAD ( Song et al. (2025)) | CVPR 25 | 44.54 | 16.71 | 170.21 | **48.63** |
| WoTE ( Li et al. (2025b)) | ICCV 25 | 61.71 | 31.36 | - | - |
| DiffAD ( Wang et al. (2025b)) | arXiv 25 | 67.92 | 38.64 | - | - |
| Orion ( Jiang et al. (2025)) | ICCV 25 | 77.74 | 56.62 | 151.48 | 17.38 |
| DriveMoE ( Yang et al. (2025)) | arXiv 25 | 74.22 | 48.64 | 175.96 | 15.31 |
| AutoVLA ( Zhou et al. (2025)) | NeurIPS 25 | 78.84 | 57.73 | 146.93 | 39.33 |
| SimLingo ( Renz et al. (2025)) | CVPR 25 | 85.07 | 67.27 | **259.23** | 33.67 |
| DE-Driver (**Ours**) | - | **86.73** | **67.73** | 245.99 | 17.61 |

Table 2: Multi-Ability Test Results (%) on the Bench2Drive Benchmark.

| Method | Merging | Overtaking | Emergency Brake | Give Way | Traffic Sign | **Mean** |
|---|---|---|---|---|---|---|
| TCP-traj* ( Wu et al. (2022)) | 8.89 | 24.29 | 51.67 | 40.00 | 46.28 | 34.22 |
| AD-MLP ( Zhai et al. (2023)) | 0.00 | 0.00 | 0.00 | 0.00 | 4.35 | 0.87 |
| VAD ( Jiang et al. (2023)) | 8.11 | 24.44 | 18.64 | 20.00 | 19.15 | 18.07 |
| UniAD-Base ( Hu et al. (2023)) | 14.10 | 17.78 | 21.67 | 10.00 | 14.21 | 15.55 |
| ThinkTwice* ( Jia et al. (2023b)) | 27.38 | 18.42 | 35.82 | 50.00 | 54.23 | 37.17 |
| DriveAdapter* ( Jia et al. (2023a)) | 28.82 | 26.38 | 48.76 | 50.00 | 56.43 | 42.08 |
| DriveTransformer ( Jia et al. (2025)) | 17.57 | 35.00 | 48.36 | 40.00 | 52.10 | 38.60 |
| DiffAD ( Wang et al. (2025b)) | 30.00 | 35.55 | 46.66 | 40.00 | 46.32 | 38.79 |
| Orion ( Jiang et al. (2025)) | 25.00 | **71.11** | 78.33 | 30.00 | 69.15 | 54.72 |
| DriveMoE ( Yang et al. (2025)) | 34.67 | 40.00 | 65.45 | 40.00 | 59.44 | 47.91 |
| SimLingo ( Renz et al. (2025)) | 54.01 | 57.04 | **88.33** | 53.33 | **82.45** | 67.03 |
| DE-Driver (**Ours**) | **60.00** | 56.25 | 83.33 | **60.00** | 79.44 | **67.81** |

the number of activated parameters ranges from 0.95 to 1.11 billion during inference. DE-Driver achieves a remarkable reduction in inference time, particularly in the LLM decoding stage. Our model's average LLM decode latency is only **657.8ms**, a nearly **44% improvement** over Sim-Lingo's 1176.9ms. This substantial speed-up is a direct consequence of our adaptive framework; the scene-aware router correctly identifies simpler scenarios, bypassing the powerful but slow deliberative expert and CoT generation. This allows DE-Driver to resolve the critical trade-off between performance and latency, paving the way for more practical deployment of advanced VLA models.

## 4.3 QUALITATIVE RESULTS

To further validate the performance of DE-Driver, we present a side-by-side comparison with Sim-Lingo in two challenging scenarios from the Bench2Drive benchmark, as shown in Figure 4. In the scenario (a), the ego-vehicle should turn left at the fork but was blocked by the white car in the left lane. At the beginning (Frame 10), both models made similar decisions, but DE-Driver operated reactively, generating a concise action ("...Executing...") with a latency of only **370 ms**, less than half of SimLingo's 947 ms. As the vehicle approached the fork, DE-Driver activated more deliberative experts and generated longer CoT ("Decelerate to drive with the target speed..."), to handle this difficult scenario. Although taking longer, DE-Driver turned left into the fork successfully. In contrast, Simlingo with rigid reasoning process, generated "...Accelerate..." commands repeatedly, resulting

Table 3: Model Parameters and Inference Latency of Each Modules of VLA.

| Method | Param. (B) | Vision | | LLM | | | Action Head | | Average latency |
|---|---|---|---|---|---|---|---|---|---|
| | | Param. (M) | Latency (ms) | Param. (M) | Prefill Latency | Decode Latency | Param. (M) | Latency (ms) | |
| Orion | 7.51 | 334 | 136.9 | 6755 | 75.9 | 6892.1 | 134.33 | 163.2 | 7268.1 |
| SimLingo | 1.27 | 308 | 15.4 | 961 | 45.1 | 1176.9 | 0.85 | 0.2 | 1237.6 |
| Reactive Expert | 0.95 | 308 | 15.4 | 641 | **36.6** | 710.5 | 0.85 | 0.2 | 762.7 |
| DE-Driver (**Ours**) | 1.43 (0.95-1.11) | 308 | **15.4** | 801 | 41.2 | **657.8** | 0.85 | **0.2** | 714.6 |

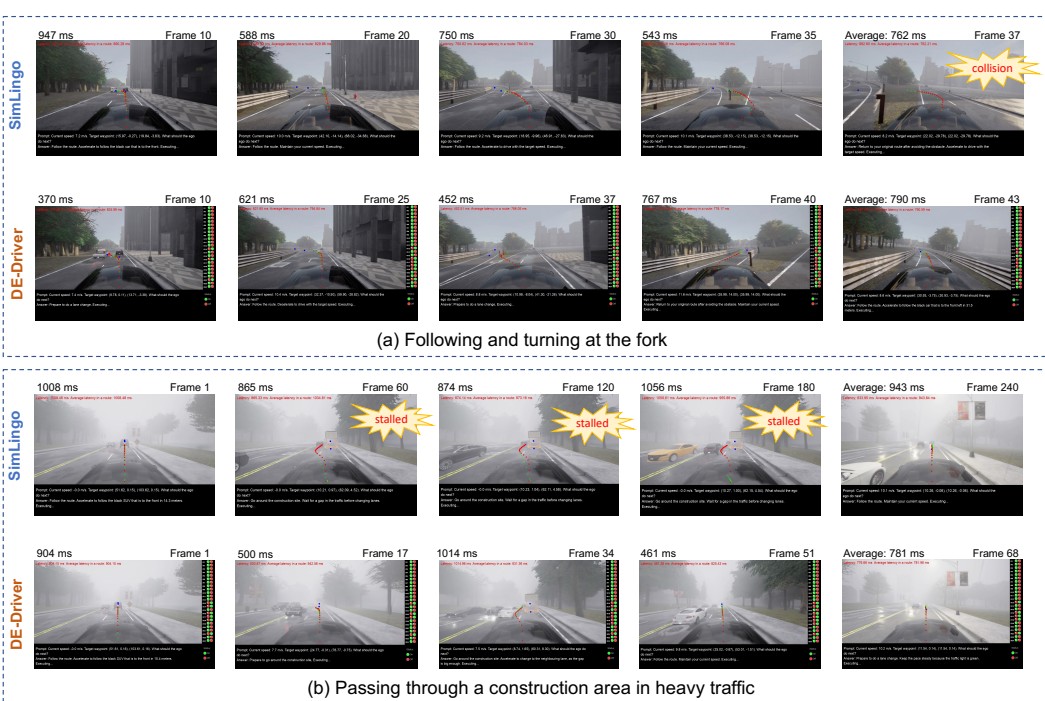

Figure 4: Qualitative Comparison. (a) SimLingo: failed due to collision, DE-Driver: successful. (b) SimLingo: successful after 240 frames, DE-Driver: successful after 68 frames. The green and red circles on the right side of DE-Driver screenshots represent the dynamical selection between reactive and deliberative experts at each blocks.

in a collision at Frame 37. The scenario (b) highlights DE-Drvier's superior efficiency when navigating a construction zone. While Simlingo becomes overly cautious, repeatedly generating "Wait for a gap in the traffic..." and remaining stationary for an extended period (taking 240 frames to pass), DE-Driver demonstrated better tactical awareness. It identifies an early opportunity to merge, executing the bypass swiftly and completing the segment in just **69 frames**-more than three times faster. Crucially, the visualization of expert usage (the indicator bar on the right of DE-Driver's frames) provides insight into its internal mechanism. The model relies on the lightweight reactive expert for simple phases, but dynamically engaged the powerful deliberative expert for the critical merge decision, striking an optimal balance between rapid execution and careful reasoning. This adaptive behavior is the key to its significant advantage in both effectiveness and efficiency.

Meanwhile, Figure 5 provides a more systematic view of how the deliberative expert is activated across different types of driving scenarios and how this relates to the end-to-end latency. The ego experienced a sequence of driving maneuvers at the traffic intersection and most deliberative expert are activated at the challenging scenarios(*e.g.*, 'turning', 'encountering oncoming vehicles', 'changing lane' or 'merging in or out intersection')and disabled at the simple scenarios (*e.g.*, 'wait-

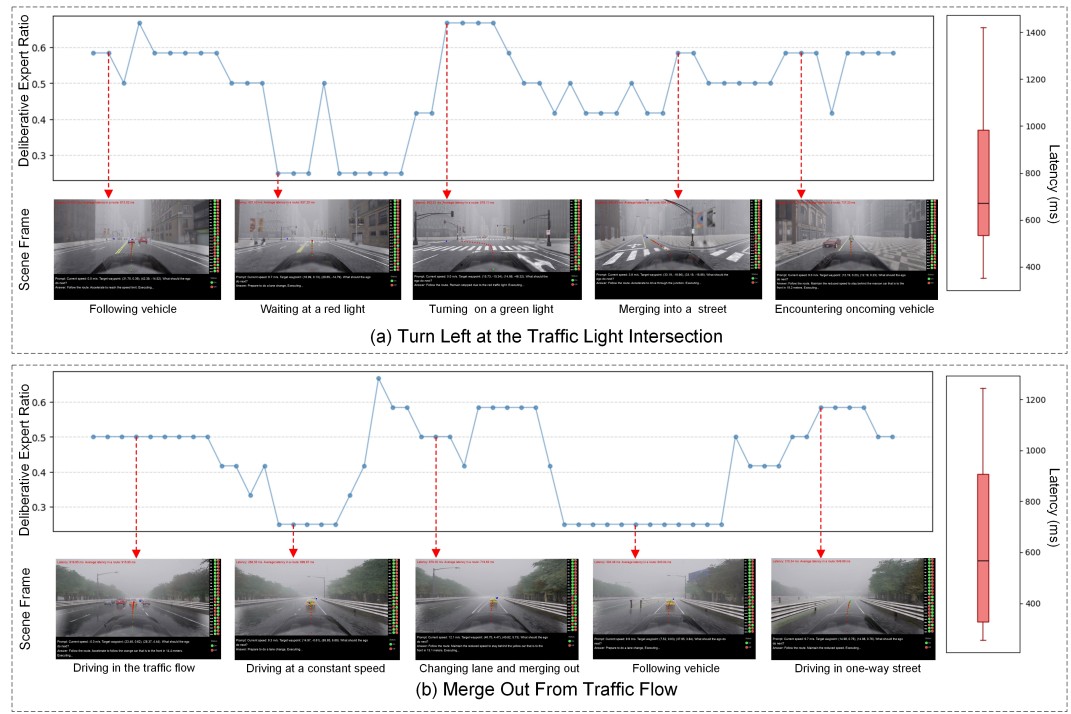

Figure 5: Visualization of Deliberative Expert Activation and System Latency.

Table 4: Ablation Studies on the Core Components of DE-Driver.

(a) Effect of Meta-Architecture.

| Setting | DS | SR(%) | Mean(%) |
|---------|----|----|------|
| Baseline (SimLingo) | 81.96 | 65 | 63.16 |
| + DE | 79.00 | 60 | 68.83 |
| + PES (Ours) | **87.34** | **70** | **76.00** |

(b) Effect of Expert Load Loss Components.

| Setting | DS | SR(%) | Mean(%) |
|---------|----|----|------|
| $l_s$ | 71.11 | 50 | 58.33 |
| + $l_{\mathrm{agent}}$ | 85.04 | 65 | **81.33** |
| + $l_{\mathrm{text}}$ (Ours) | **87.34** | **70** | 76.00 |

(c) Ablation of Reactive Expert Training.

| Setting | DS | SR(%) | Mean(%) |
|---------|----|----|------|
| Baseline (Our Deliberative) | 81.96 | 65 | 63.16 |
| + Reduce Param. | 75.79 | 55 | 64.50 |
| + Pruning (Stage 1) | 81.71 | 65 | **73.00** |
| + Distill (Stage 2: Our Reactive Expert) | **83.82** | 60 | 70.50 |
| + Stage 3 (DE-Driver) | 87.34 | 70 | 76.00 |

ing moment at a red light', 'Following route or vehicle in constant speed'). Furthermore, scenario latency demonstrates that allocating different experts can improves effectiveness and efficiency.

## 4.4 ABLATION STUDIES

We conducted a series of ablation studies to dissect the DE-Driver framework and validate the effectiveness of its key components. Note that constrained by the extensive evaluation time required by Bench2drive, we conducted our ablation experiments exclusively on a curated set of 20 challenging routes for comparative analysis. The results are summarized in Table 4 and 5.

**Effect of Core Architecture and Training Strategy:** Table 4 (a) investigates the synergy between our dual-expert architecture and the proposed training strategy. We start from SimLingo (81.96 DS). Introducing our dual-expert (DE) modules without the specialized Progressive Expert Specialization

Table 5: Ablation Studies on Chain-of-Thought (CoT) and Long-Tail Scenarios.

(a) Effect of CoT.

| Setting | DS | SR(%) | Mean(%) |
|---------|-----|-------|---------|
| w/o CoT | 80.51 | 55 | 69.00 |
| w/ CoT | **87.34** | **70** | **76.00** |

(b) Results on the Top-20 Long-Tail Scenarios.

| Setting | DS | SR(%) | Latency |
|---------|-----|-------|---------|
| SimLingo | 69.35 | 45.0 | 1010.48 |
| DE-Driver | 87.80 | 70.0 | 805.52 |

(PES) strategy leads to a performance degradation (79.00 DS). This highlights that merely equipping the model with two experts is insufficient; without proper functional differentiation, the router struggles to make optimal choices, leading to confused predictions. However, when the full PES strategy is employed, the model's performance dramatically improves to a **87.34 DS** and **70% SR**. This unequivocally demonstrates that our three-stage PES pipeline is critical for cultivating specialized experts and training a precise router, forming the foundation of DE-Driver's success.

**Analysis of Router Loss Functions:** We analyze the impact of different supervisory signals for our scene-aware router in Table 4 (b). Using only the basic scene-difficulty label ($l_s$) yields a modest performance (71.11 DS), indicating that a simple binary signal is too coarse. Incorporating the agent proportion loss ($l_{\text{agent}}$) provides a substantial boost to 85.04 DS, confirming that the presence of other vehicles is a powerful heuristic for scene complexity. Finally, adding the CoT length proportion loss ($l_{\text{text}}$) further refines the router's behavior by correlating expert selection with reasoning depth, leading to our best performance. This shows that a combination of explicit and implicit complexity signals is essential for optimal routing.

**Formulation of the Reactive Expert:** Table 4 (c) validates our methodology for creating a competent yet lightweight reactive expert. Compared to the baseline, naively reducing FFN parameters ('+ Reduce Param.') severely degrades performance to 75.79 DS, as it discards vital knowledge of LLM. In contrast, using structured pruning from the full deliberative expert ('+Pruning') effectively preserves performance (81.71 DS). Crucially, applying knowledge distillation ('+Distill') allows the pruned expert to not only recover but also refine its capabilities, reaching a strong 83.82 DS. This confirms that our distillation process is essential for creating a reactive expert that is both efficient and highly capable, enabling it to handle the majority of driving scenarios effectively.

**Effect of CoT:** To validate the necessity of the chain-of-thought, we conducted an ablation study by disabling the CoT generation, as Table 5 (a). The results indicate a significant performance degradation without explicit reasoning, with the driving score dropping from 87.34 to 80.51 and the success rate falling from 70% to 55%. This sharp decline confirms that CoT can often offer more high-level semantic logic for complex end-to-end driving tasks.

**Results of Long-Tail Scenarios:** To further demonstrate the efficacy of our adaptive mechanism, we categorized the evaluation routes into hard (long-tail) and easy scenarios. As shown in Table 5 (b), DE-Driver achieves a remarkable performance boost, improving the DS from 69.35 to 87.80 in challenging long-tail scenarios. This suggests that our specialized Deliberative Expert, free from the interference of simple patterns during the PES training, learns more robust reasoning capabilities for corner cases than a monolithic model. More results are illustrated in Appendix A.7.

## 4.5 CONCLUSION

We propose DE-Driver, a novel dual-expert VLA model that adaptively switches specialized experts and reduces unnecessary reasoning chain for high-efficiency autonomous driving. A dual-expert module is introduced to degrade active parameters and prefilling time by assign deliberative and reactive experts by the scene-aware router. A progressive expert specialization strategy is employed to align the sematic space of two expert and optimize the decode latency for generating reasoning chain. DE-Driver achieves state-of-the-art closed-loop performance while maintaining a low inference latency. Though promising, the dual-expert system still face the below limitations: High-frequency oscillations in expert selection between adjacent time steps lead to discontinuous decision-making, reducing the smoothness of driving and comfort. In future work, we will focus on introducing a temporal processing mechanism to make smoother expert selections by considering the history of states and actions, thereby enhancing both behavioral continuity and overall efficiency.

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

# A APPENDIX

## A.1 DETAILS OF SCENE-AWARE ROUTER

As described in Section 3.2, our scene-aware router can capture the contextual semantics of different scenarios. Its schematic diagram is illustrated in Figure 6. During the training stage, both experts accept all vision tokens to extract action features. The router then analyzes the vision features to decide which expert's output should be weighted for use in the next layer. Notably, the Gumbel Argmax is crucial for enabling gradient backpropagation through this selection process during training. The inference process is described as Eq. (4), based on the logits produced by the router, only the expert with the highest score is activated to perform inference, while the other remains idle.

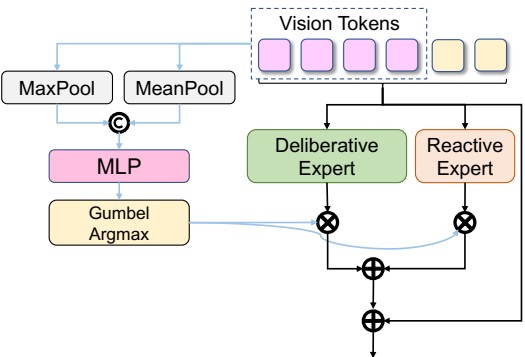

Figure 6: Details of Scene-Aware Router

## A.2 MORE TRAINING DETAILS

Our model employs an autoregressive Transformer architecture to process multi-modal sequences. The input to the Transformer is a concatenated sequence of tokens representing visual, language, and action information, structured as [Image, Text, Action]. To ensure that the model's predictions are causal (i.e., a prediction for a token at a given position can only depend on previous tokens), we apply a specific attention mask during both training and inference. The structure of this mask is illustrated in Figure 7. The masking strategy is defined as follows:

**Image Tokens:** Image tokens can only attend to other image tokens. This allows the model to build a comprehensive visual representation by performing self-attention exclusively within the visual modality. They are masked from attending to any subsequent text or action tokens.

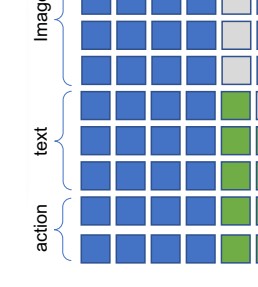

Figure 7: Attention map

**Text Tokens:** Text tokens can attend to all preceding image tokens and all preceding (including the current) text tokens. This cross-modal attention is crucial for grounding the language instruction in the visual context. However, text tokens are prevented from attending to any action tokens, ensuring the model does not "see the future" action sequence when processing the command.

**Action Tokens:** Action tokens can attend to all preceding tokens, including all image tokens, all text tokens, and all previous action tokens. This enables the model to generate the current action based on the full context: the visual scene (Image), the language command (Text), and other waypoints (Action).

This structured attention mask enforces a strict autoregressive generation process for the reasoning of CoT, which is fundamental to our model's ability to context-aware action plans. The model is trained using a standard cross-entropy loss to predict the next token, where the loss is computed only on the text. All action tokens interact with other tokens in parallel, improving the inference speed.

## A.3 RULE-BASED FILTER

To reduce unnecessary CoT in simple scenarios, we design a rule-based filter to reconstruct the dataset of SimLingo (Renz et al. (2025)) into a mixed dataset $D_{mix}$ with and without CoT based on scenario complexity. This allows the VLA to handle challenging cases that benefit from CoT. We formulate rules to quantify complexity along three axes: vehicle jerk, environmental density, and maneuver type.

**Vehicle Jerk.** As a measure of motion smoothness and comfort, we calculate the mean jerk of the annotated trajectory. Given waypoints $p = ((x_1, y_1), ..., (x_t, y_t))$ sampled at intervals $\Delta t$, the jerk magnitude $J(t)$ is approximated using a finite difference method:

$$J(t) = \sqrt{j_x(t)^2 + j_y(t)^2} \tag{10}$$

where $j_x(t) = \frac{x_{t+2} - 2x_{t+1} + 2x_{t-1} - x_{t-2}}{2(\Delta t)^3}$ (and analogously for $j_y$). The final metric is the mean of all valid $J(t)$ values.

**Environmental Complexity.** This metric quantifies the density of relevant surrounding agents, including the vehicle, pedestrian, and traffic sign. In the annotation, an object is considered relevant if it is in the ego-vehicle's frontal view and within a class-specific distance and angle threshold. We compute a normalized environmental complexity score $y_{agent} \in [0, 1]$:

$$y_{\text{agent}} = \frac{\min(N_{\text{relevant}}, N_{\text{max}})}{N_{\text{max}}} \tag{11}$$

where $N_{\text{relevant}}$ is the count of relevant objects and $N_{\text{max}}$ is a scaling factor (set to 5).

**Driving Behavior Analysis.** Based on annotated driving waypoints, we classify a maneuver as complex if it involves significant acceleration, deceleration, a lane change, or a turn, which usually represent that the driver responded due to some special circumstances. Each behavior is detected if a corresponding metric (e.g., acceleration magnitude, lateral path deviation, heading change rate) exceeds a predefined threshold. The final Driving Behavior Complexity $S_{\text{maneuver}}$ is a boolean indicator:

$$S_{\text{maneuver}} = B_{\text{accel}} \vee B_{\text{decel}} \vee B_{\text{lane\_change}} \vee B_{\text{turn}} \tag{12}$$

where each $B$ term is a boolean flag for the corresponding detected behavior.

Finally, our rule-based filter can classify the scenarios by the jerk $J$, environmental complexity score $y_{\text{agent}}$, or maneuver $S_{\text{maneuver}}$ into the simple or complex label, as Eq. (13). Meanwhile, we eliminate the explanatory thought chain (*e.g.*, "reduce speed due to a red car in the front") but maintain the high-level action instruction (*e.g.*, "turn right").

$$S_{\text{diff}} = J \vee (y_{\text{agent}} == 1) \vee B_{\text{maneuver}} \tag{13}$$

## A.4 ROUTE SELECTION ON ABLATION STUDY

We select 20 challenging routes to evaluate the performance of DE-Driver, including senarios Accident(2534), AccidentTwoWays(3410, 25845), HardBreakRoute (24330), HazardAtSideLane (1792), HighwayCutIn (2286), HighwayExit (23658), InterurbanActorFlow (24098), InterurbanAdvancedActorFlow (23708), InvadingTurn (3575), NonSignalizedJunctionLeftTurn (2084), NonSignalizedJunctionRightTurn (3670), ParkedObstacle (25318), SignalizedJunctionLeftTurnEnterFlow (28330), SignalizedJunctionRightTurn (26956), StaticCutIn (26396, 26405), T_Junction (28035), VanillaSignalizedTurnEncounterGreenLight (14909), VehicleTurningRoutePedestrian (3737).

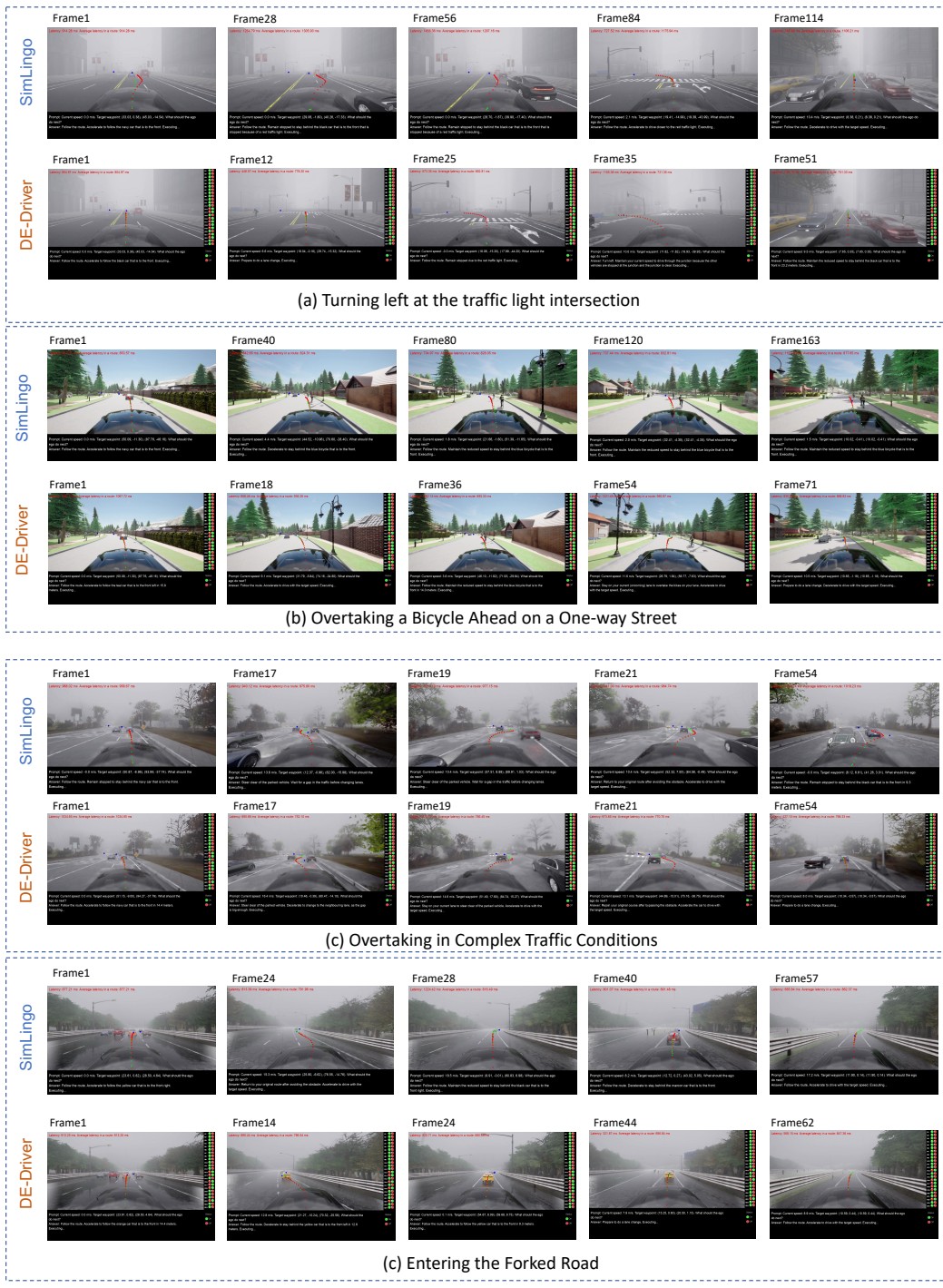

Figure 8: More Visualizations both SimLingo and DE-Driver

## A.5    MORE VISUALIZATIONS

Additional visualizations have been introduced to further demonstrate the efficacy of our proposed method, as shown in Figure 8. Our method exhibits superior performance over SimLingo, particularly in overtaking and merging maneuvers, while concurrently achieving lower system latency.

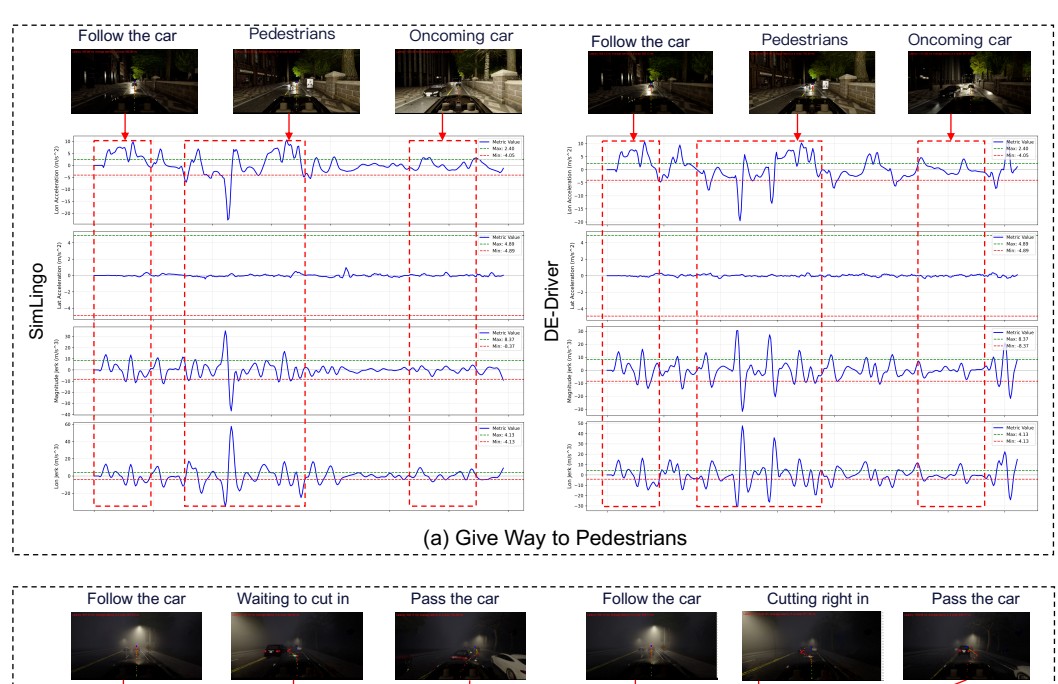

(a) Give Way to Pedestrians

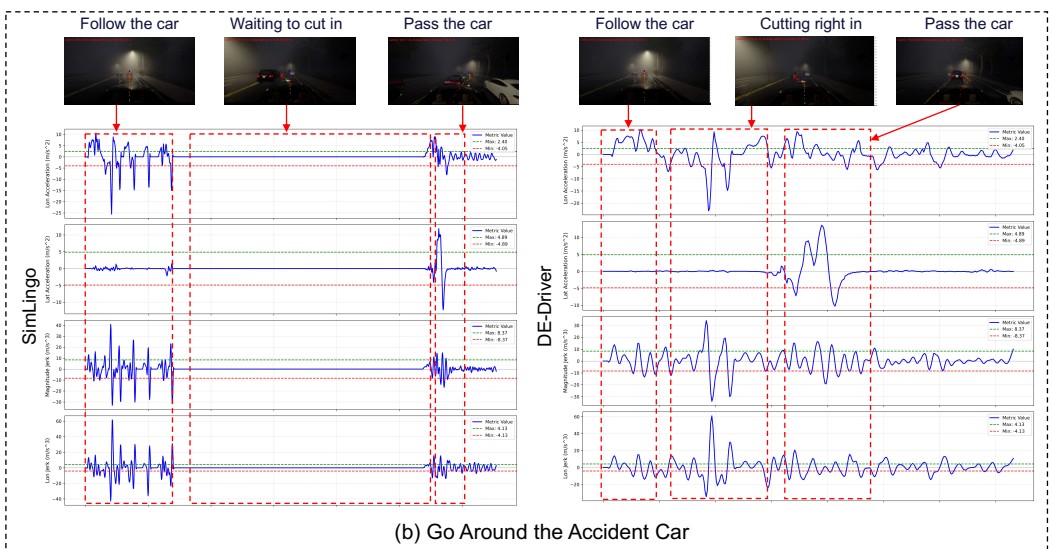

(b) Go Around the Accident Car

Figure 9: Comparison of Driving Acceleration and Jerk Over Time.

### A.6 COMFORT ANALYSIS

We obtain a lower comfort score compared to some baselines in Table 1. We conduct observation and analysis from driving style and expert switching.

**Driving Behavior Analysis:** Figure 9 displays the driving acceleration and jerk on same route between SimLingo (Renz et al. (2025)) and DE-Driving. The results demonstrate that the driving behavior of our method is more aggressive, resulting in greater fluctuations in acceleration and jerk compared to the SimLingo, as shown in Figure 9 (a). Meanwhile, SimLingo's driving style is more conservative, preferring to trade longer waiting times for greater comfort and higher success rates, as illustrated in Figure 9 (b).

**Effect of Expert Switching:** We introduced a post-processing strategy termed Slide Average Window Filter (SAWF)with a window size of 5 to smooth the router selection. As reported in Table 6, applying SAWF effectively suppresses high-frequency expert switching. The application of SAWF improves the comfort score from 21.17 to 25.92, confirming that stabilizing the expert choice directly leads to smoother driving trajectories. However, this smoothness comes at a cost. The Success Rate (SR) drops from 70.0% to 65.0%, and the Driving Score (DS) decreases from 87.34 to 86.26.

Table 6: Results of Post-processing in the Expert Switching. SAWF represents a slide average window filter in the expert switch processing.

| Setting | DS↓ | SR(%)↓ | comfort↑ | Efficiency↓ |
|---------|------|--------|----------|-------------|
| w/o SAWF | 87.34 | 70.0 | 21.17 | 268.46 |
| w/ SAWF | 86.26 | 65.0 | 25.92 | 263.69 |

Table 7: Performance Comparison on the Different Scenarios.

| Setting | Hard40 | | Easy40 | | Hard20 | | Easy20 | |
|---------|--------|-------|--------|-------|--------|-------|--------|-------|
| | DS | SR(%) | DS | SR(%) | DS | SR(%) | DS | SR(%) |
| SimLingo | 80.43 | 60.0 | 89.56 | 72.5 | 69.35 | 45.0 | 93.75 | 80.0 |
| DE-Driver | 86.95 | 67.5 | 89.11 | 67.5 | 87.80 | 70.0 | 93.73 | 75.0 |

This indicates that the "jitter" is partly a side effect of the model's high sensitivity and rapid responsiveness to environmental changes. Forcefully smoothing the expert selection introduces a lag in cognitive mode switching (e.g., failing to switch to Deliberative mode instantly when a sudden hazard appears), thereby compromising safety.

## A.7 MORE RESULTS IN LONG-TAIL SCENARIOS

**Score Scene Difficulty:** To further validate the effectiveness of DE-Driver in handling long-tail scenarios, we conduct a detailed analysis by categorizing the evaluation routes into distinct difficulty levels. we partition the dataset into two groups: *Hard* (long-tail) and *Easy* scenarios. Concretely, we use Qwen3-VL-8B-Instruct to assess the difficulty of 220 test routes across three aspects: environment, road conditions, and agent interaction. (1) Environment score $S_{env}$: The environment dimension considers factors such as weather and time of day (e.g., daytime or nighttime). (2) Road score $S_{road}$: The road dimension evaluates road complexity, including the presence of obstacles, construction zones, traffic signals, and similar elements. (3) Agent score $S_{agent}$: The agent dimension assesses trajectory complexity, as well as the difficulty and frequency of interactions with other vehicles, bicycles, and pedestrians. Each dimension is scored from 1(very easy) to 10 (very hard). Then a weighted average final score $S_{final}$ is calculated as the final difficulty score.

$$S_{final} = 0.25 * S_{env} + 0.25 * S_{road} + 0.5 * S_{agent} \tag{14}$$

The top 20 routes and the top 40 routes are marked as Hard20 and Hard40 respectively, while the lowest-scoring routes are marked as Easy20 and Easy40.

**Experimental Results:** We report experimental results on the Hard40, Easy40, Hard20, and Easy20 subsets to validate marginal performance in long-tail situations in Table 7. On the Hard40 set, DE-Driver achieves an improvement of 6.52 DS and 7.5% SR compared to SimLingo. Crucially, on the most challenging *Hard20* subset, DE-Driver substantially surpassing SimLingo by 18.45 DS and 25.0% SR. The above results and Table 4 demonstrate that our multi-stage training strategy preserves the semantic knowledge of LLM while adapting it to autonomous driving tasks, leading to significantly improved performance on corner cases for our dual-expert architecture. Interestingly, DE-Driver's advantage is less pronounced in simple scenarios, as the ratio of deliberative expert decreases.

Figure 10 illustrates that DE-Driver maintains lower average inference latency than SimLingo. Furthermore, the latency gap between DE-Driver and SimLingo is larger in simple scenarios but narrows in complex ones. The observation demonstrate that DE-Drive can dynamically allocates the powerful deliberative expert and generates CoT, thereby balancing the traditional tension between driving scores and efficiency in long-tail autonomous driving.

## A.8 PRUNING RATIO ABLATION STUDY

We further analyze how the pruning ratio used to construct the reactive expert affects the trade-off between driving performance and efficiency. As shown in Table 8, we evaluate three pruning ratios

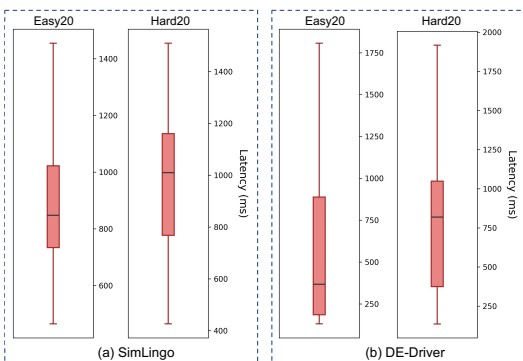

Figure 10: Latency Comparison Between SimLingo and DE-Driver on the Hard20 Scenario.

Table 8: Effect of Different Pruning Ratio

| Pruning ratio | DS | SR(%) | Latency |
|---|---|---|---|
| 30% | 84.92 | 60.0 | 659.52 |
| 50% | 87.34 | 70.0 | 714.60 |
| 70% | 87.80 | 65.0 | 741.82 |

(30%, 50%, 70%). When we prune 50% of the FFN intermediate channels to build the reactive expert, DE-Driver achieves a 87.34DS, 70.0%SR and 714.60ms latency. At 30% pruning ratio, DS and SR drops by 2.42 and 10%, respectively. The 50% pruning ratio maintains performance comparable to the 70% pruning ratio, while offering better efficiency. Therefore, we adopt a 50% pruning ratio as the default setting in this work.

