# OpenReview forum: "Deliberation Meets Reaction: A Dual-Expert VLA framework for Autonomous Driving"
_ICLR.cc/2026/Conference — Submitted to ICLR 2026_

### Official Review · Reviewer_JnUT · 2025-10-21

**Soundness:** 2
**Presentation:** 3
**Contribution:** 2
**Rating:** 4
**Confidence:** 4

**Summary:**

This paper proposes DE-Driver, a novel vision-language-action (VLA) model featuring a dual-expert architecture that adaptively switches between specialized experts and prunes unnecessary reasoning chains to achieve high-efficiency autonomous driving. DE-Driver demonstrates competitive performance across multiple efficiency metrics and significantly reduces inference time.

**Strengths:**

1/ The paper introduces a Dual-Expert (DE) module that dynamically switches between a lightweight reactive expert and a more powerful deliberative expert, enabling context-aware computation.

2/ A Progressive Expert Specialization (PES) strategy is proposed to enforce distinct expert roles and stabilize router training, selectively skipping the generation of unnecessary chains of thought (CoT).

3/ DE-Driver gets a new state-of-the-art on the Bench2Drive benchmark.

**Weaknesses:**

1/ The dual-expert design—comprising a lightweight reactive expert and a powerful deliberative expert—resembles a classic teacher-student framework, which is not uncommon in model compression or scenario-adaptive systems.

2/ The reliability of the Scene-Aware Router remains a concern: misjudgments by the router could lead to the omission of essential CoT reasoning in complex scenarios, potentially compromising decision quality and safety.

3/ While the DE module reduces latency, it increases overall system complexity. The overall number of parameters is more than before.

**Questions:**

1/ Beyond improving inference efficiency, does skipping unnecessary CoT generation have any adverse effects on model training stability or final performance?

2/ Since the experiments rely on data collected from the CARLA simulator, which may not fully capture the complexity of real-world driving. How likely is it to skip CoT reasoning would diminish significantly in more diverse and challenging real-world scenarios?

3/ Could the authors provide statistics on the proportion of decisions handled by each expert? Moreover, is this allocation ratio stable across different traffic conditions or scene types, or does it vary significantly?

4/ Does the Scene-Aware Router incorporate mechanisms such as a cooldown period (i.e., minimum dwell time per expert) or hysteresis to prevent rapid switching that might destabilize vehicle control and degrade ride comfort—as alluded to in the paper’s discussion of comfort-aware improvements?

5/ The Structured Pruning strategy is mentioned but not described in detail. Could the authors provide a quantitative analysis linking the reduction in model parameters to concrete gains in latency and performance?

---

> ### Author Response · Authors · 2025-11-21
>
> Thank you very much for the valuable comments and helpful suggestions on our paper. I hope my answers clarify the motivation and novelty and dispel your doubts. Our itemized responses to your comments are listed as follows:
> ***
> **Response to Weakness 1**:
> To further clarify our contribution, (1)we have added additional statements to better articulate our motivation and highlight the novelty of our approach in Section 2.2 of the revised paper.
>
> (2)We have conducted several experiments to validate the effectiveness and efficiency of the DE-Driver in Section 4.3, Section 4.4 and Appendix A6-A8 in the revised manuscript.
> ***
> **Response to Weakness 2**:  We gratefully appreciate for your valuable suggestion. We have conducted an additional experiment to validate the  reliability of the Scene-Aware Router. We first verify the necessary of CoT. We compare our scene-aware router with CoT against  one without CoT in Table 5(a) in the revised manuscript. The results shown that removing CoT causes a performance drop by 6.83 DS, 15% SR, and  7 Mean. Therefore,  these results validate that our Scene-Aware Router correctly identifies situations requiring reasoning and does not compromise safety.
>
> ***
> **Response to Weakness 3**: Thanks for your valuable comments. We have  clarified this calculation in Para. 3 of Section 4.2 and Table 3 in the revised paper: "Our model adopts an MoE architecture with a total of 1.43 billion parameters, the number of activated parameters ranges from 0.95 to 1.11 billion during inference"
>
> ***
> **Response to Question 1**: Thanks for your insightful comment. Through extensive experiments in Table 7 and Figure 10 in the revised manuscript, we observe two key effects for adaptive CoT generation: (1) In simple scenarios, our method achieves lower latency with no significant improvement in driving scores; (2) In complex scenarios, DE-Driver delivers better driving scores, but the gap of latency improvement narrows.
>
> ***
> **Response to Question 2**: Thank you very much for your insightful questions. We agree that real-world scenarios are significantly more diverse and challenging, thus we argue the importance of CoT becomes even more critical.
>
> We have conducted several experiments to validate the necessity of CoT and advantage of adaptive CoT generation in Table 5 in the revised manuscript. Therefore, we argue that CoT is more critical for complex real-world driving. However, generating CoT only when necessary is more efficient for end-to-end autonomous driving.
>
> ***
> **Response to Question 3**: Special thanks for your valuable comments. We have added Figure 5 to show the ratio of expert switching, scene frame, and latency box plot in the revised paper. Driving routes typically vary in complexity, such as urban roads, highways, and driving scenarios are also different within a single route, such as traffic lights or car-following. Thus, the ratio of deliberative expert must be adaptively adjusted over time. In the Figure 5 of the revised paper, the ego experienced a sequence of driving maneuvers at the traffic intersection and most deliberative expert are activated at the challenging scenarios and disabled at the simple scenarios.
> ***
> **Response to Question4**: Thank you very much for your valuable advice. We have added an additional experiments to analysis the cause in  Appendix A.6 of the revised manuscript.
>
> (1)Different driving styles also affect comfort. We display the longitudinal and lateral acceleration and jerk during driving in Figure 9 of the revised manuscript. The results show that our method is more aggressive and reacts quickly, whereas SimLingo drives more smoothly and tends to stop and wait in many emergency situations, which consumes a significant amount of time and thus contributes to their higher comfort score.
>
> (2)We employ a sliding window filter to reduce expert switching. As shown in the table below or Table 6 of the revised paper, the results indicate that while simple post-processing methods can improve comfort(21.17->25.92), but they degrade other performance metrics, which is undesirable. Indeed, expert switching does lead to reduced comfort, which remains a key direction for our future work.
>
> ***
> **Response to Question 5**:  We gratefully appreciate for your valuable suggestion. We have add experiment results of different pruning ratios in the Table 8 of the revised manuscript. We evaluate three pruning ratios (30%, 50%, 70%). When we prune 50% of the FFN intermediate channels to build the reactive expert, DE-Driver achieves a 87.34DS, 70.0%SR and 714.60ms latency. At 30% pruning ratio, DS and SR drops by  2.42 and 10%, respectively. The 50% pruning ratio maintains performance comparable to the 70% pruning ratio, while offering better efficiency. Therefore, we adopt a 50% pruning ratio as the default setting in this work.
> ***
>
> Again, thanks for your professional review of our paper. Your constructive suggestions have
> definitely helped to improve our paper.

---

### Official Review · Reviewer_UYDC · 2025-10-29

**Soundness:** 2
**Presentation:** 3
**Contribution:** 2
**Rating:** 4
**Confidence:** 3

**Summary:**

This work identifies latency as a key bottleneck for applying VLA models in autonomous driving. To mitigate this, a novel approach is proposed, which combines structured pruning, knowledge distillation, and an adaptive expert router. The core idea is to enable the model to automatically alternate between experts in response to different scenarios. This strategy is designed to decrease inference latency to near real-time levels, without a significant loss in decision-making performance.

**Strengths:**

This paper attempts to address a critical challenge in applying VLA (Vision-Language-Action) models to autonomous driving inference tasks: enabling real-time and accurate inference. Furthermore, the authors conduct their experiments in a closed-loop policy environment, which enhances the practical significance of the findings.

**Weaknesses:**

The experimental evaluation does not sufficiently address the core problem outlined in the paper. The experiments are primarily focused on driving performance, failing to adequately demonstrate the proposed method's improvements in terms of latency. The paper would be significantly strengthened by presenting an analysis of the trade-off between inference performance (e.g., accuracy) and inference latency, which would provide a more comprehensive and convincing evaluation.

**Questions:**

- In the caption of Figure 1, the subfigure indicator `(a)` appears to be incorrectly written as `(1)`. Please correct this typo.
- There seems to be an inconsistency regarding the model size of DE-Driver reported in Table 3. The table shows that DE-Driver has fewer parameters than SimLingo. However, the description accompanying Figure 3 suggests that DE-Driver utilizes two experts for inference, with the larger expert being architecturally equivalent to SimLingo in terms of parameter count. Logically, the total parameters of DE-Driver (the sum of both experts) should be greater than, or at the very least equal to, that of SimLingo. The authors should clarify this discrepancy.
- The analysis of inference latency could be significantly strengthened with the following additions: a) It would be more informative to report the total end-to-end inference latency of the model, rather than only providing a breakdown of latency for each module. b) To better illustrate the overall latency distribution, the authors could present the per-frame latency data using a box plot or a violin plot. This would offer readers clearer insights into the model's performance consistency and variance. c) A crucial experiment would be to investigate the relationship between scene complexity and inference latency. Does the latency of DE-Driver converge towards that of SimLingo as scene complexity increases, or does it remain slightly lower? Including such targeted experiments on inference latency would substantially enrich the paper's contributions.

---

> ### Author Response · Authors · 2025-11-21
>
> Thank you very much for your valuable processing of our paper and helpful comments on our paper. Our responses to your comments are listed as follows:
> ***
> **Response to Weakness**: Thanks for your comment. We have added some additional experimental results to validate effectiveness of DE-Drive in Figure 5, Table 5(b), and Figure 10 of the revised manuscript. These results demonstrate that our adaptive reasoning  mechanism achieves a balance between inference accuracy and efficiency. When the deliberative expert is selected more frequently, the model exhibits stronger reasoning capabilities and generates more CoT data, thereby improving trajectory accuracy. In contrast, for simpler scenarios, the deliberative expert is activated less often, reducing complex reasoning outputs and consequently lowering system latency.
> ***
> **Response to Question 1** : We sincerely thank the reviewer for pointing out this typo. The subfigure indicator in the caption of Figure 1 has been corrected from “(1)” to “(a)” as suggested.
>
> ***
> **Response to Question 2**: We thank the reviewer for this insightful question. We have  clarified this calculation in Para. 3 of Section 4.2 and Table 3 in the revised paper: "Our model adopts an MoE architecture with a total of 1.43 billion parameters, the number of activated parameters ranges from 0.95 to 1.11 billion during inference"
>
> Concretely, the reported parameter count reflects our specific hybrid architecture described in Section 3.1 and Eq. (2), rather than a simple duplication of experts across all layers. As detailed in Section 4.1, for the 24-layer LLM backbone, we set M=12: the first 12 layers use a fixed, lightweight Reactive Expert (structurally pruned) to minimize latency, while only the last 12 layers employ the Dual-Expert MoE structure. This means the heavy Deliberative Expert is excluded from the first half of the network, making the maximum number of activated parameters lower than the full SimLingo baseline.
> ***
> **Response to Question 3**: Thanks for your comprehensive suggestions.
>
> **Response to Question 3 a)**: We have added the average latency  in Table 3 in the revised paper.
> | Method           | Average latency (ms) |
> |------------------|----------------------|
> | Orion            | 7268.1               |
> | SimLingo         | 1237.6               |
> | Reactive Expert  | 762.7                |
> | DE-Drive         | 714.6                |
>
>
> **Response to Question 3 b)**: We add the Figure 5 to show the ratio of epert switching, scene frame, and latency box plot in the revised paper. The ego experienced a sequence of driving maneuvers at the traffic intersection and most deliberative expert are activated at the challenging scenarios(turning,  encountering oncoming vehicles, changing lane or merging in or out intersection )and disabled at the simple scenarios (waiting moment at a red light, Following route or vehicle in constant speed). Furthermore, scenario latency demonstrates that allocating different experts can improves effectiveness and efficiency.
>
> **Response to Question 3 c)**: We have conducted specific experiments to test the relationship between scene complexity and inference latency. In the top 20 hardest scenarios (Hard20), DE-Driver achieved a Driving Score (DS) of 87.80, a Success Rate (SR) of 70.0%, and 805.52 ms average latency, significantly outperforming SimLingo's DS of 69.35, SR of 45.0%, and 1010.48 ms. Meanwhile, In the top 20 easiest scenarios (Easy20), the our method achieves performance comparable to SimLingo, but with superior latency.
> | Setting     | Hard20 DS | Hard20 SR(%) | Hard20 Latency | Easy20 DS | Easy20 SR(%) | Easy20 Latency |
> |-------------|-----------|--------------|----------------|-----------|--------------|----------------|
> | SimLingo    | 69.35     | 45.0         | 1010.48        | 93.75     | 80.0         | 890.03         |
> | DE-Driver   | 87.80     | 70.0         | 805.52         | 93.73     | 75.0         | 542.94         |
>
> Furthermore, a detialed latency distribution is shown in Figure 10 in the revised manuscrip, showing that DE-Driver maintains lower average inference latency than SimLingo even on the Hard20 scenarios, confirming that our adaptive routing does not sacrifice speed for complexity. The inference latency reduction is larger under simple conditions, but converges toward baseline performance in complex scenes.
> ***
> Again, thanks for your professional review of our paper. Your insightful comments have definitely helped to improve the quality of our paper.

---

> > ### Comment · Reviewer_UYDC · 2025-11-27
> >
> > Thanks to the kindly response, I will raise my score.

---

> > > ### Author Response · Authors · 2025-11-27
> > > **Sincere Thanks to Reviewer UYDC**
> > >
> > > Dear Reviewer UYDC,
> > >
> > > We sincerely thank you for your time and support. Your suggestions have been invaluable in refining this paper!
> > >
> > > Best regards,
> > >
> > > The Authors of Paper #23338

---

### Official Review · Reviewer_2qVr · 2025-10-30

**Soundness:** 3
**Presentation:** 3
**Contribution:** 3
**Rating:** 4
**Confidence:** 3

**Summary:**

This paper introduces DE-Driver, a Vision-Language-Action (VLA) model that aims to improve the efficiency of autonomous driving systems while maintaining high performance. DE-Driver incorporates a dual-expert system with two specialized experts: a reactive expert for handling simple scenarios and a deliberative expert for more complex situations requiring detailed reasoning. The system dynamically switches between these experts using a scene-aware router based on the difficulty of the driving task. Additionally, the authors introduce a Progressive Expert Specialization (PES) strategy for training these experts and improving their performance. DE-Driver achieves state-of-the-art performance on the Bench2Drive benchmark, demonstrating both high driving performance and reduced inference latency. The model outperforms existing methods in terms of computational efficiency without sacrificing decision-making quality.

**Strengths:**

1. A dual-expert system with adaptive switching between reactive and deliberative experts is a novel contribution to the field of autonomous driving. This approach addresses the significant trade-off between computational cost and performance by selectively activating the necessary expertise based on scenario complexity.
2. The Progressive Expert Specialization (PES) strategy for training the reactive and deliberative experts is a clever method for balancing efficiency and performance. It enhances the model’s ability to perform under diverse conditions and allows for both specialized pruning and knowledge distillation to create a highly efficient model.
3. DE-Driver outperforms several baselines on the Bench2Drive benchmark, achieving state-of-the-art driving scores while reducing computational costs. Its strong performance in complex scenarios (e.g., merging, overtaking) further demonstrates its robustness.

**Weaknesses:**

1. While the paper compares DE-Driver to a number of other methods, the comparison with models using Mixture of Experts (MoE) or similar adaptive reasoning models is limited. A more comprehensive comparison with models that dynamically adjust computation based on task complexity (e.g., DriveMoE) would provide a clearer picture of the model's relative strengths.
2. The reactive expert, while lightweight and efficient, may struggle with more complex reasoning tasks, as evidenced by some performance degradation when it is used alone. The pruning process, while effective, still introduces some limitations in generalization to more challenging scenarios.
3. The evaluation is based on closed-loop benchmarks, which are useful but may not fully reflect real-world variability. Testing on more diverse or real-world datasets could further validate the model's robustness and generalizability across a wider range of driving environments.

**Questions:**

1. Have you considered testing DE-Driver in real-world driving environments? How does it perform in terms of both safety and latency in these environments?
2. You mention that expert switching sometimes leads to discontinuous decision-making, affecting comfort. Have you considered implementing a temporal processing mechanism that smooths out transitions between experts, such as incorporating memory-based methods or using a sliding window to consider past actions?
3. The paper shows DE-Driver's strong performance in complex tasks like overtaking and merging. How does the model generalize to long-tail tasks or situations not seen during training (e.g., rare road events, unusual pedestrian behavior)?

---

> ### Author Response · Authors · 2025-11-21
>
> Thank you very much for your valuable processing of our paper and helpful comments on our paper. Our responses to your comments are listed as follow:
> ***
> **Response to Weakness 1**:  Thanks for your insightful suggestion. We have expanded our evaluation to include additional experiments on adaptive reasoning, as presented in Figure 5 and Table 5(a) of the revised manuscript.  Firstly, DriveMoE learns different driving skills with different experts, which is conceptually different from our work. Moreover, there are no other similar MoE methods that have been applied to autonomous driving, so we can only conduct more comparisons with the baseline model. Figure 5 and Table 5(a) of the revised manuscript demonstrate that our adaptive reasoning  mechanism achieves a balance between inference accuracy and efficiency. When the deliberative expert is selected more frequently, the model exhibits stronger reasoning capabilities and generates more CoT data, thereby improving trajectory accuracy. In contrast, for simpler scenarios, the deliberative expert is activated less often, reducing complex reasoning outputs and consequently lowering system latency.
> ***
>
> **Response to Question1 and Weakness3**: Thanks for your valuable suggestion on real-world testing. Currently, Bench2Drive remains the most authoritative and widely adopted closed-loop evaluation benchmark. Its interactive nature  enables a more realistic simulation of real-world driving scenarios, where different driving decisions trigger varying behaviors from other vehicles and pedestrians. However, due to time constraints, we have not yet had the opportunity to systematically extend CoT data annotation, adapt our training framework, or establish full closed-loop metrics on real-world datasets. The proposed method in this paper is specifically designed for our team’s practical deployment pipeline on our autonomous vehicle, and we are currently conducting sim-to-real experiment using real-world driving data. We look forward to presenting our next work, which will be based on real-world data.
>
> ***
>
> **Response to Question 2**: Thanks for your constructive suggestion on temporal smoothing. We have added an additional results to analysis the cause in  Appendix A.6 of the revised manuscript.
>
> (1)Different driving styles also affect comfort. We display the longitudinal and lateral acceleration and jerk during driving in Figure 9 of the revised manuscript. The results show that our method is more aggressive and reacts quickly, whereas SimLingo drives more smoothly and tends to stop and wait in many emergency situations, which consumes a significant amount of time and thus contributes to their higher comfort score.
>
> (2)We employ a sliding window filter to reduce expert switching. As shown in the table below or Table 6 of the revised paper, the results indicate that while simple post-processing methods can improve comfort(21.17->25.92), but they degrade other performance metrics, which is undesirable. Indeed, expert switching does lead to reduced comfort, which remains a key direction for our future work.
> | Setting         | DS↓     | SR(%)↓ | Comfortness↑ | Efficiency↓ |
> |-----------------|---------|--------|--------------|-------------|
> | Without SAWF    | 87.34   | 70.0   | 21.17        | 268.46      |
> | with SAWF       | 86.26   | 65.0   | 25.92        | 263.69      |
>
>
> ***
> **Response to Question 3 and Weakness 2**: Thanks for your important question about generalization to long-tail tasks. We have expanded our evaluation to include additional experiments on long-tail scenarios, as presented in Table 5(b) and Appendix A.7 of the revised manuscript. Evaluating the performance on long-tail scenarios is indeed central to our work, and we have conducted specific experiments to address this. In the top 20 hardest scenarios (Hard20), DE-Driver achieved a Driving Score (DS) of 87.80 and a Success Rate (SR) of 70.0%, significantly outperforming SimLingo's DS of 69.35 and SR of 45.0%.
>
> | Setting     | Hard20 DS | Hard20 SR(%) | Hard20 Latency | Easy20 DS | Easy20 SR(%) | Easy20 Latency |
> |-------------|-----------|--------------|----------------|-----------|--------------|----------------|
> | SimLingo    | 69.35     | 45.0         | 1010.48        | 93.75     | 80.0         | 890.03         |
> | DE-Driver   | 87.80     | 70.0         | 805.52         | 93.73     | 75.0         | 542.94         |
>
> Furthermore, other sections of the paper also support our focus on and the model's effectiveness in long-tail scenarios, Table 2 in the revised paper shows that DE-Driver achieves the top performance in several complex interactive tasks, such as "Merging" and "Give Way," further validating its capability in handling challenging traffic situations.
> ***
> Again, thanks for your professional review of our paper. Your insightful suggestions have definitely helped to improve our paper.

---

### Official Review · Reviewer_cnJ7 · 2025-11-01

**Soundness:** 3
**Presentation:** 3
**Contribution:** 3
**Rating:** 6
**Confidence:** 4

**Summary:**

DE-Driver targets a critical, practical challenge in VLA-based autonomous driving—balancing performance and inference efficiency—and offers an innovative dual-expert solution aligned with human driving intuition. The paper’s strengths lie in its well-motivated adaptive framework, structured PES training strategy, and comprehensive closed-loop evaluation on Bench2Drive, which validates both SOTA driving performance and significant latency reductions. However, several limitations undermine the work’s depth and generalizability: the scene-aware router’s complexity assessment lacks dynamic interaction modeling (e.g., unpredictable pedestrian behavior), the CoT generation logic for the deliberative expert is underspecified, long-tail scenario robustness is not explicitly evaluated, and comparisons to recent efficiency-focused VLA models (e.g., FastDriveVLA) are missing. Additionally, the trade-off between expert switching and driving comfort (acknowledged as low comfortness scores) lacks actionable mitigation details. Addressing these gaps, via expanded evaluations, clarified technical mechanisms, and more inclusive baselines, would strengthen DE-Driver’s contribution to efficient, deployable VLA systems.

**Strengths:**

The paper directly addresses a critical pain point of existing VLA models: their prohibitive computational cost (due to billions of parameters and mandatory CoT reasoning) makes them incompatible with resource-constrained onboard hardware. DE-Driver’s dual-expert design, leveraging a lightweight reactive expert for ~80% of simple scenarios (per human driving analogies) and a deliberative expert only when necessary, resolves this by adapting computation to scenario complexity rather than using a one-size-fits-all VLA backbone. This focus on efficiency without performance loss aligns with real-world autonomous driving deployment needs, distinguishing it from works that prioritize performance alone.

**Weaknesses:**

CoT generation logic for the deliberative expert is underspecified. While the paper claims the deliberative expert generates CoT only when necessary, it provides no details.

Missing comparisons to recent efficiency-focused VLA baselines, DE-Driver claims to advance “efficient VLA design,” but it does not compare to recent works with similar goals.

The paper frames VLA models’ strength as handling long-tail scenarios, but DE-Driver’s performance on these is untested.

The paper acknowledges that DE-Driver has lower comfort scores (17.61) than baselines like SimLingo (33.67) and attributes this to expert switching. However, no details are provided on why switching causes discomfort (e.g., abrupt changes in acceleration, conflicting trajectory suggestions from adjacent layers). No mitigation strategies are proposed beyond a vague future temporal processing mechanism. For example, a simple smoothing layer between expert outputs or a switching hysteresis (avoiding frequent toggles between experts) could reduce oscillations, but these are not discussed.

**Questions:**

CoT generation logic for the deliberative expert is underspecified. While the paper claims the deliberative expert generates CoT only when necessary, it provides no details.

Missing comparisons to recent efficiency-focused VLA baselines, DE-Driver claims to advance “efficient VLA design,” but it does not compare to recent works with similar goals.

The paper frames VLA models’ strength as handling long-tail scenarios, but DE-Driver’s performance on these is untested.

The paper acknowledges that DE-Driver has lower comfort scores (17.61) than baselines like SimLingo (33.67) and attributes this to expert switching. However, no details are provided on why switching causes discomfort (e.g., abrupt changes in acceleration, conflicting trajectory suggestions from adjacent layers). No mitigation strategies are proposed beyond a vague future temporal processing mechanism. For example, a simple smoothing layer between expert outputs or a switching hysteresis (avoiding frequent toggles between experts) could reduce oscillations, but these are not discussed.

---

> ### Author Response · Authors · 2025-11-21
>
> Thank you very much for your valuable processing of our paper and helpful comments on our paper.  Our responses to your comments are listed as follows:
> ***
> *Question1: CoT generation logic for the deliberative expert is underspecified. While the paper claims the deliberative expert generates CoT only when necessary, it provides no details.*
>
> **Response to Question 1:**  Thanks for your  constructive comment.  We have clarified the CoT details in Para. 5, Section 3.1 and Para. 4, Section 3.3 in the revised paper.
> ***
> *Question2: Missing comparisons to recent efficiency-focused VLA baselines, DE-Driver claims to advance “efficient VLA design,” but it does not compare to recent works with similar goals.*
>
> **Response to Question 2**: Thanks for your insightful comment. Most VLA methods focus more on improving driving scores, and we observe that  FastDriveVLA is a efficient frame and focuses on reducing spatial redundancy by pruning background visual tokens, which primarily optimizes the prefill stage. Its prefill time is 51 ms on H800 GPUs, and our method is 41.2 ms on 4090 GPU. Although DE-Driver achieves faster inference speed, FastDriveVLA has not been open-sourced, and we are unable to test it on the same hardware. We consider this unfair. Therefore, we have not included these comparisons in the paper.
> ***
> *Question3: The paper frames VLA models’ strength as handling long-tail scenarios, but DE-Driver’s performance on these is untested.*
> **Response to Question 3**: Thanks for your valuable comment. We have expanded our evaluation to include additional experiments on long-tail scenarios, as presented in Table 5(b) and Appendix A.7 of the revised manuscript. Evaluating the performance on long-tail scenarios is indeed central to our work, and we have conducted specific experiments to address this. In the top 20 hardest scenarios (Hard20), DE-Driver achieved a Driving Score (DS) of 87.80 and a Success Rate (SR) of 70.0%, significantly outperforming SimLingo's DS of 69.35 and SR of 45.0%.
>
> | Setting     | Hard20 DS | Hard20 SR(%) | Hard20 Latency | Easy20 DS | Easy20 SR(%) | Easy20 Latency |
> |-------------|-----------|--------------|----------------|-----------|--------------|----------------|
> | SimLingo    | 69.35     | 45.0         | 1010.48        | 93.75     | 80.0         | 890.03         |
> | DE-Driver   | 87.80     | 70.0         | 805.52         | 93.73     | 75.0         | 542.94         |
>
> Furthermore, other sections of the paper also support our focus on and the model's effectiveness in long-tail scenarios, Table 2 in the paper shows that DE-Driver achieves the top performance in several complex interactive tasks, such as "Merging" and "Give Way," further validating its capability in handling challenging traffic situations.
> ***
> *Question4: The paper acknowledges that DE-Driver has lower comfort scores (17.61) than baselines like SimLingo (33.67) and attributes this to expert switching. However, no details are provided on why switching causes discomfort (e.g., abrupt changes in acceleration, conflicting trajectory suggestions from adjacent layers). No mitigation strategies are proposed beyond a vague future temporal processing mechanism. For example, a simple smoothing layer between expert outputs or a switching hysteresis (avoiding frequent toggles between experts) could reduce oscillations, but these are not discussed.*
>
> **Response to Question 4**: Thanks for your constructive question. We have added an additional results to observe the cause in  Appendix A.6 of the revised manuscript.
> (1)Different driving styles also affect comfort. We display the longitudinal and lateral acceleration and jerk during driving in Figure 9 of the revised manuscript. The results show that our method is more aggressive and reacts quickly, whereas SimLingo drives more smoothly and tends to stop and wait in many emergency situations, which consumes a significant amount of time and thus contributes to their higher comfort score.
>
> (2)We employ a sliding window filter to reduce expert switching. As shown in the table below or Table 6 of the revised paper, the results indicate that while simple post-processing methods can improve comfort(21.17->25.92), but they degrade other performance metrics, which is undesirable. Indeed, expert switching does lead to reduced comfort, which remains a key direction for our future work.
> | Setting         | DS↓     | SR(%)↓ | Comfortness↑ | Efficiency↓ |
> |-----------------|---------|--------|--------------|-------------|
> | Without SAWF    | 87.34   | 70.0   | 21.17        | 268.46      |
> | with SAWF       | 86.26   | 65.0   | 25.92        | 263.69      |
> ***
> Again, we sincerely appreciate the time and effort invested by the reviewer in evaluating our manuscript.

---

### Author Response · Authors · 2025-11-25
**Summary of the response**

## Summary of the response

We sincerely thank all reviewers for their valuable time and constructive feedback, which have greatly improved the quality of our paper.

On one hand , we have uploaded a video to the **supplementary materials** to verify the effectiveness of our proposed method.

On another hand, we summarize the questions that reviewers have focused on:
***
> ***Question1: Reasons for Low Comfort Scores***

**Reply:** We have added an additional results to analysis the reason in  Appendix A.6 of the revised manuscript.
(1)Effect of different driving styles. We display the longitudinal and lateral acceleration and jerk during driving in Figure 9 of the revised manuscript. The results show that our method is more aggressive and reacts quickly, whereas SimLingo drives more smoothly and tends to stop and wait in many emergency situations, which consumes a significant amount of time and thus contributes to their higher comfort score.

(2) Effect of expert switching. As shown in the table below or Table 6 of the revised paper, the results indicate that while simple post-processing methods can improve comfort(21.17->25.92), but they degrade other performance metrics, which is undesirable. Indeed, expert switching does lead to reduced comfort, which remains a key direction for our future work.

| Setting       | DS↓   | SR(%)↓ | Comfort↑ | Efficiency↓ |
|---------------|-------|--------|----------|-------------|
| Without SAWF  | 87.34 | 70.0   | 21.17    | 268.46      |
| with SAWF     | 86.26 | 65.0   | 25.92    | 263.69      |

***
> ***Question2: Visualization and Latency Statistics Regarding Expert Switching***

**Reply:** We add the Figure 5 to show the ratio of expert switching, scene frame, and latency box plot in the revised paper. The ego experienced a sequence of driving maneuvers at the traffic intersection and most deliberative expert are activated at the challenging scenarios( 'turning',  'encountering oncoming vehicles', 'changing lane or merging in or out intersection') and disabled at the simple scenarios ('waiting moment at a red light', 'Following route or vehicle in constant speed'). Furthermore, scenario latency demonstrates that allocating different experts can improves effectiveness and efficiency.

***
> ***Question3: Reliability under Long-Tail Scenarios***

**Reply:*** We have expanded our evaluation to include additional experiments on long-tail scenarios, as presented in below Table or Table 5(b) and Appendix A.7 of the revised manuscript. Evaluating the performance on long-tail scenarios is indeed central to our work, and we have conducted specific experiments to address this. In the top 20 hardest scenarios (Hard20), DE-Driver achieved a Driving Score (DS) of 87.80 and a Success Rate (SR) of 70.0%, significantly outperforming SimLingo's DS of 69.35 and SR of 45.0%.

| Setting     | Hard20              |   Hard20                |       Hard20            | Easy20              |       Easy20            |       Easy20            |
|-------------|---------------------|-------------------|-------------------|---------------------|-------------------|-------------------|
|             | DS                  | SR(%)             | Latency           | DS                  | SR(%)             | Latency           |
| SimLingo    | 69.35               | 45.0              | 1010.48           | 93.75               | 80.0              | 890.03            |
| DE-Driver   | 87.80               | 70.0              | 805.52            | 93.73               | 75.0              | 542.94            |

Furthermore, other sections of the paper also support our focus on and the model's effectiveness in long-tail scenarios, Table 2 in the revised paper shows that DE-Driver achieves the top performance in several complex interactive tasks,  further validating its capability in handling challenging traffic situations.

***

> ***Question4: Reliability of CoT and Scene-Aware Router***

**Reply:** We have conducted an additional experiment to validate the  reliability of the CoT and Scene-Aware Router. We compare our scene-aware router with CoT against  one without CoT in below Table or Table 5(a) in the revised manuscript. The results shown that removing CoT causes a performance drop by 6.83 DS, 15% SR, and  7 Mean. Therefore, CoT reasoning is absolutely essential for achieving high performance and safety. Meanwhile, these results demonstrate that the Scene-Aware Router is reliable to select deliberate expert and generate CoT at suitable moments. If the router frequently misjudged complex scenes and omitted CoT, our model's performance would inevitably degrade towards the poor results of the "w/o CoT".  Thus, these results validate that our Scene-Aware Router correctly identifies situations requiring reasoning and does not compromise safety.

| Setting   | DS    | SR(%) | Mean(%) |
|-----------|-------|-------|---------|
| w/o CoT   | 80.51 | 55    | 69.00   |
| with CoT  | 87.34 | 70    | 76.00   |

---

### Author Response · Authors · 2025-12-01
**Summary Reply to AC and all Reviewers**

We've been notified by the ICLR Program Chairs of the recent security incident involving reviewer anonymity on the OpenReview platform. We sincerely appreciate the organizers’ efforts to maintain fairness in the peer-review process and  the AC’s intervention at a critical moment.

Throughout the process, all authors strictly followed the anonymous review policy. Before the incident occurred, we had already carefully prepared point‑by‑point responses to all reviewer comments, which resulted in a score improvement from Reviewer UYDC. Although further communication with the reviewers is no longer possible, their feedback played a crucial role in improving our paper. Therefore, we would like to report the revisions we have made to the AC as a gesture of respect for the reviewers’ contributions.
***
## Summary of Key Responses to Reviewer Concerns

### 1. ***Reasons for Low Comfort Scores (Reviewer cnJ7 &  2qVr & JnUT)***

We have added an additional results to analysis the reason in  **Appendix A.6** of the revised manuscript.

(1) **Effect of different driving styles**. We display the longitudinal and lateral acceleration and jerk during driving in **Figure 9** of the revised manuscript. The results show that our method is more aggressive and reacts quickly, whereas SimLingo drives more smoothly and tends to stop and wait in many emergency situations, which consumes a significant amount of time and thus contributes to their higher comfort score.

(2) **Effect of expert switching**. As shown in the **Table 6** of the revised paper, the results indicate that while simple post-processing methods can improve comfort (21.17->25.92), but they degrade other performance metrics, which is undesirable.

| Setting       | DS↓   | SR(%)↓ | Comfortness↑ | Efficiency↓ |
|---------------|-------|--------|--------------|-------------|
| Without SAWF  | 87.34 | 70.0   | 21.17        | 268.46      |
| with SAWF     | 86.26 | 65.0   | 25.92        | 263.69      |

***
### 2. ***Visualization and Efficiency Comparison Regarding Expert Switching (Reviewer cnJ7 & UYDC & JnUT)***

(1) We have uploaded a video to the **supplementary materials** to visualize the driving scenario, latency, and expert switching of our proposed method and SimLingo.

(2) We have added the **Figure 5** to show the ratio of expert switching and latency box plot in the revised paper.

(3) We have added the average latency in **Table 3** in the revised paper.

| Method           | Average latency (ms) |
|------------------|----------------------|
| Orion            | 7268.1               |
| SimLingo         | 1237.6               |
| Reactive Expert  | 762.7                |
| DE-Drive         | 714.6                |

***

### 3. ***Reliability Under Long-Tail Scenarios (Reviewer cnJ7 & 2qVr)***

We have expanded our evaluation to include additional experiments on long-tail scenarios, as presented in **Table 5(b)** and **Appendix A.7** of the revised manuscript.

| Setting     | Hard20             |     Hard20               |    Hard20                | Easy20             |  Easy20                  |  Easy20                  |
|-------------|--------------------|--------------------|--------------------|--------------------|--------------------|--------------------|
|             | DS                 | SR(%)              | Latency            | DS                 | SR(%)              | Latency            |
| SimLingo    | 69.35              | 45.0               | 1010.48            | 93.75              | 80.0               | 890.03             |
| DE-Driver   | 87.80              | 70.0               | 805.52             | 93.73              | 75.0               | 542.94             |

***

### 4. ***Reliability of CoT and Scene-Aware Router (Reviewer UYDC & JnUT)***

We have conducted an additional experiment to validate the  reliability of the CoT and Scene-Aware Router in **Table 5(a)**  in the revised paper.

| Setting   | DS     | SR(%) | Mean(%) |
|-----------|-------:|------:|--------:|
| w/o CoT   | 80.51  |    55 |   69.00 |
| with CoT  | 87.34  |    70 |   76.00 |

***

### 5. ***Model Parameter Interpretation and Hyperparameter Ablation (Reviewer UYDC & JnUT)***
(1) **Model Parameter Interpretation**. We have  clarified model parameter in **Para. 3 of Section 4.2 and Table 3** in the revised paper.

(2) **Hyperparameter Ablation**. We have add experiment results of different pruning ratios in the  **Table 8** of the revised manuscript.
| Pruning ratio | DS     | SR(%) | Latency |
|---------------|-------:|------:|--------:|
| 30%           | 84.92  | 60.0  | 659.52  |
| 50%           | 87.34  | 70.0  | 714.60  |
| 70%           | 87.80  | 65.0  | 741.82  |

***

Again, we sincerely thank all reviewers and the ACs for their time, thoughtful feedback, and commitment to scientific rigor. Despite the recent unfortunate platform issues, we remain fully dedicated to maintaining transparent, responsible, and constructive scholarly communication.

---

### Meta-Review · Area_Chair_uj4A · 2025-12-27

**Summary:**

This paper proposes DE-Driver, an adaptive dual-expert Vision-Language-Action (VLA) framework that dynamically routes inputs between a lightweight reactive expert and a reasoning-heavy deliberative expert to balance driving performance with computational efficiency. However, several limitations remain:

1. Comfort Gap and Switching Instability: Although the proposed sliding window filter partially mitigates control oscillations, the driving comfort score remains significantly lower than SOTA (e.g., SimLingo), and the framework lacks a systematic architectural solution to resolve the inherent discontinuity caused by expert switching.

2. Deployment Feasibility and Efficiency Analysis: The inference latency remains prohibitively high for practical on-vehicle deployment, even in the simplest scenarios (approx. 543 ms) ; furthermore, the study lacks a comprehensive trade-off curve (performance vs. efficiency) to demonstrate the method's viability under the strict real-time constraints required for actual deployment.

3. Generalization to Real-World Domains: The evaluation relies exclusively on the CARLA simulator without validation on real-world datasets , raising concerns that the current heuristic rules for scenario complexity and router training may be overfitted to the simulation environment and fail to generalize to open-world driving tasks.

**Reviewer Concerns:**

1. Driving Comfort and Expert Switching Stability A primary concern shared by multiple reviewers (cnJ7, 2qVr, JnUT) was the model's low "comfortness" scores (17.61) compared to the SimLingo baseline (33.67). Reviewers questioned whether the dynamic switching between experts caused discontinuous decision-making.

2. Router and CoT Mechanism Verification Concerns were raised regarding the opacity of the CoT generation logic and the potential safety risks if the scene-aware router fails to activate the deliberative expert.

3. Inference Efficiency and Latency The inference latency remains a significant concern, creating a substantial gap before practical deployment. The fundamental trade-off between performance and efficiency has not been thoroughly resolved.

4. The absence of real-world dataset experiments and the expert switching classification rules being specifically designed for CARLA simulation environments.

**Reviewer Scores:**

*   **Reviewer cnJ7**: 6 -> 6. The concerns about comfort have not been fundamentally resolved.
*   **Reviewer 2qVr**: 4 -> 4. The reviewer expected the authors to test on real-world datasets rather than relying entirely on simulators, and the authors did not provide a strong response to this point.
*   **Reviewer UYDC**: 4 -> 6. The reviewer's initial concerns have been partially addressed.
*   **Reviewer JnUT**: 4 -> 4. The absence of experiments on real-world datasets remains unaddressed. The non-smooth transitions caused by strategy switching have not been systematically resolved.

---

### Decision · Program_Chairs · 2026-01-26

Reject